# ACTIVE: Offline Reinforcement Learning via Adaptive Imitation and In-sample $V$-Ensemble

**Tianyuan Chen**[1,3,6]   **Ronglong Cai**[1,3]   **Faguo Wu**[1,3,4,6*]   **Xiao Zhang**[2,3,4,5,6*]

[1]School of Artificial Intelligence, Beihang University
[2]School of Mathematical Sciences, Beihang University
[3]Key Laboratory of Mathematics, Informatics and Behavioral Semantics, MoE, Beihang University
[4]Beijing Advanced Innovation Center for Future Blockchain and Privacy Computing, Beihang University
[5]Hangzhou International Innovation Institute of Beihang University
[6]Zhongguancun Laboratory

## Abstract

Offline reinforcement learning (RL) aims to learn from static datasets and thus faces the challenge of value estimation errors for out-of-distribution actions. The in-sample learning scheme addresses this issue by performing implicit TD backups that does not query the values of unseen actions. However, pre-existing in-sample value learning and policy extraction methods suffer from over-regularization, limiting their performance on suboptimal or compositional datasets. In this paper, we analyze key factors in in-sample learning that might potentially hinder the use of a milder constraint. We propose Actor-Critic with Temperature adjustment and In-sample Value Ensemble (ACTIVE), a novel in-sample offline RL algorithm that leverages an ensemble of $V$-functions for critic training and adaptively adjusts the constraint level using dual gradient descent. We theoretically show that the $V$-ensemble suppresses the accumulation of initial value errors, thereby mitigating overestimation. Our experiments on the D4RL benchmarks demonstrate that ACTIVE alleviates overfitting of value functions and outperforms existing in-sample methods in terms of learning stability and policy optimality.

## 1 Introduction

Reinforcement learning (RL) has achieved significant success in various sequential decision-making tasks (Mnih et al., 2015; Silver et al., 2016). However, a notable drawback of online RL is the requirement for a huge number of continual environmental interactions, which might be impractical since real-world interactions can be costly or dangerous. Offline RL aims to learn from static datasets without further interactions (Levine et al., 2020), which in principle makes it possible to exploit large previously collected datasets. In practice, the distributional shift between the data-collecting policy (also known as the behavior policy) and the learned policy poses significant challenges for offline RL algorithms. Improving the policy beyond the the dataset level requires querying the values of actions sampled from the policy, which are likely to be out-of-distribution. These out-of-distribution actions can produce highly overestimated $Q$-values (Kumar et al., 2019; Fujimoto et al., 2019), which can be further propagated with bootstrapping, resulting in catastrophic inaccuracies in value estimation.

To address the overestimation problem, prior offline RL methods typically add pessimism to the learning objective either by constraining the policy to be close to the behavior policy (Wu et al., 2019; Fujimoto et al., 2019; Kumar et al., 2019; Peng et al., 2019; Nair et al., 2020; Fujimoto & Gu, 2021) or by regularizing the value function to make pessimistic predictions for out-of-distribution actions (Kumar et al., 2020; Kostrikov et al., 2021; An et al., 2021; Yang et al., 2022). More recently, implicit TD backups have been proposed in IQL (Kostrikov et al., 2022) that completely avoid querying values of unseen actions by approximating the in-sample maximum of $Q$-values using expectile regression. However, when the dataset is skewed towards suboptimal policies or mixed with lower-quality data, prior in-sample learning methods could suffer from performance drop due to over-regularization (Xiao et al., 2023).

---

*Corresponding Authors. Emails: {faguo,xiao.zh}@buaa.edu.cn.

In this paper, we aim at improving the overall performance of IQL-style in-sample offline RL methods (Kostrikov et al., 2022; Xu et al., 2023; Garg et al., 2023). Empirically, we find that in implicit TD backups, $V(s)$ could easily overfit the initial error of $Q(s, a)$, causing catastrophic overestimation and performance degradation. This phenomenon prevents the use of a larger expectile $\tau$ or a lower level of implicit regularization $\alpha$ (Xu et al., 2023). Based on this observation, we propose an in-sample value learning scheme based on a $V$-ensemble that mitigates the overfitting of value functions, thus allowing for less regularization. The in-sample $V$-ensemble captures *in-sample epistemic uncertainty*, which is then used as a penalty during bootstrapping and advantage estimation. To better balance generalization and regularization in suboptimal datasets, we propose a policy extraction loss inspired by automated entropy adjustment in SAC (Haarnoja et al., 2018) that adaptively adjusts the strength of imitation based on dual gradient descent.

Our primary contribution is Actor-Critic with Temperature adjustment and In-sample Value Ensemble (ACTIVE), an ensemble-based in-sample offline RL algorithm that combines the aforementioned modifications. We conduct theoretical analysis of the output dynamics of the $Q$-function and the error suppression effect of the $V$-ensemble. To promote understanding of the in-sample epistemic uncertainty, we empirically analyze the distribution of advantage functions obtained using a single network and a $V$-ensemble on D4RL datasets (Fu et al., 2020), highlighting the need for a more discriminative advantage estimate. Our experiments on the D4RL benchmark show that ACTIVE improves upon existing in-sample methods in terms of learning stability and final policy performance.

## 2 RELATED WORK

**Model-free Offline RL.** The majority of recently proposed model-free offline RL methods rely on either policy constraints or value regularization. Policy constraints can be implemented through distance constraints towards dataset actions (Kumar et al., 2019; Fujimoto & Gu, 2021; Ran et al., 2023), or through explicit modelling of the behavior policy (Wu et al., 2019; Fujimoto et al., 2019; Siegel et al., 2020; Ghasemipour et al., 2021; Zhou et al., 2021; Wu et al., 2022). Policy constraints can also be imposed implicitly using weighted behavior cloning (Wang et al., 2018; Peng et al., 2019; Nair et al., 2020). Value regularization methods drive the critic to be pessimistic about out-of-distribution state-action pairs (Kumar et al., 2020; Kostrikov et al., 2021; An et al., 2021; Yang et al., 2022). Recently, in-sample learning was proposed in IQL (Kostrikov et al., 2022) that completely avoids querying the values of any unseen actions. Generalizations of IQL, including EQL (Garg et al., 2023) and SQL (Xu et al., 2023) have been proposed. More recently, IDQL (Hansen-Estruch et al., 2023) connects IQL to actor-critic methods through an implicit actor. Our work is based on the generalized IQL framework. In contrast, our work reveals a major difficulty in implicit TD learning and provides an ensemble-based solution that can be applied to any variant of IQL.

**Ensembles and Uncertainty Estimation.** A number of studies have successfully employed ensembles for uncertainty estimation in offline RL. Ensembles can be applied to the critic in model-free offline RL (Agarwal et al., 2020; An et al., 2021; Bai et al., 2022; Ghosh et al., 2022; Ghasemipour & Gu, 2022; Yang et al., 2022) or the dynamics models in model-based offline RL (Yu et al., 2020; Kidambi et al., 2021; Sun et al., 2023). In most of these methods, ensembles provide estimates of epistemic uncertainty (Clements et al., 2019), which are then used to form a pessimistic value prediction for *out-of-distribution* samples. Addtionally, modifications like OOD sampling in PBRL (Bai et al., 2022) and gradient diversification in EDAC (An et al., 2021) have been proposed to specifically target out-of-distribution samples. In our work, we focus on in-sample learning, which does not query the values of any unseen actions during critic training. Unlike previous approaches, the ensemble in our method captures *in-sample* epistemic uncertainty, which is used to suppress error accumulation and form a more discriminative advantage estimate for in-distribution samples.

**Over-regularization in Offline RL.** While staying close to the offline dataset limits potential overestimation and stabilizes training, recent studies have argued that these constraints can be overly conservative in certain cases (Jin et al., 2021; Buckman et al., 2021; Xie et al., 2021). In practice, it has been reported that over-regularization occurs in a variety of offline RL algorithms (Zhou et al., 2021; Kumar et al., 2022; Fu et al., 2022; Xiao et al., 2023). To balance conservatism and generalization, MCQ (Lyu et al., 2022) proposed to learn a mildly conservative $Q$-function by actively training on OOD actions. From the policy constraint perspective, PRDC (Ran et al., 2023) proposed to constrain the policy toward the nearest-neighbor state-action pair. In contrast, we aim to mitigate the over-regularization of in-sample methods by combining SAC-style automated entropy adjustment with weighted behavior cloning.

## 3 PRELIMINARIES

### 3.1 OFFLINE REINFORCEMENT LEARNING

RL is formulated as a Markov Decision Process (MDP) defined as a tuple $(\mathcal{S}, \mathcal{A}, p_0, p, r, \gamma)$, with state space $\mathcal{S}$, action space $\mathcal{A}$, initial state distribution $p_0(s)$, transition dynamics $p(s'|s, a)$, reward function $r(s, a)$ and discount factor $\gamma \in (0, 1)$. The goal of RL is to find a policy $\pi(a|s) : \mathcal{S} \times \mathcal{A} \to [0, 1]$ that maximizes the expected return $\mathbb{E}_\pi[\sum_{t=0}^\infty \gamma^t r(s_t, a_t)]$ in the MDP. In this work, we focus on offline RL, where the agent only has access to a fixed dataset $\mathcal{D} = \{(s, a, r, s')\}$ collected using a different behavior policy or potentially multiple policies. The behavior policy is denoted as $\mu(a|s)$, which is the action distribution conditioned on states observed in the dataset.

### 3.2 GENERALIZED IMPLICIT Q-LEARNING

**Implicit Q-Learning.** Instead of explicitly constraining the policy or regularizing the $Q$-function, Implicit Q-learning (IQL) (Kostrikov et al., 2022) approximates the in-sample maximum of the $Q$-function using a value network $V_\psi(s)$ and expectile regression:

$$\mathcal{L}_V(\psi) = \mathbb{E}_{(s,a)\sim\mathcal{D}}[L_2^\tau(Q_{\hat{\theta}}(s, a) - V_\psi(s))], \quad \text{where} \quad L_2^\tau(u) = |\tau - \mathbb{1}(u < 0)|u^2. \quad (1)$$

The $Q$-function can then be learned by bootstrapping off the value estimate of the next state, thus avoiding any queries to values of out-of-distribution actions:

$$\mathcal{L}_Q(\theta) = \mathbb{E}_{(s,a,s')\sim\mathcal{D}}[(r(s, a) + \gamma V_\psi(s') - Q_\theta(s, a))^2]. \quad (2)$$

For policy extraction, IQL uses Advantage Weighted Regression (AWR) (Peng et al., 2019):

$$\mathcal{L}_\pi(\phi) = \mathbb{E}_{(s,a)\sim\mathcal{D}}[\exp(\beta(Q_{\hat{\theta}}(s, a) - V_\psi(s)))\log \pi_\phi(a|s)]. \quad (3)$$

**Generalized Implicit Q-Learning.** Recently, Hansen-Estruch et al. (2023) showed that IQL can be rederived as an actor-critic method. For arbitrary convex loss $f$, define the general IQL $V$-update as:

$$V^*(s) = \underset{V(s)}{\arg\min}\,\mathbb{E}_{a\sim\mu}[f(Q(s, a) - V(s))]. \quad (4)$$

The following theorem states that $f$ corresponds to an implicit actor $\pi_{\text{imp}}(a|s)$:

**Theorem 3.1** (Hansen-Estruch et al. (2023)). *Denote $\frac{\partial f}{\partial V(s)}$ as $f'$, then for every state $s$ and convex loss function $f$ such that $f'(0) = 0$, the solution to the optimization problem defined in Equation (4) is also a solution to the optimization problem*

$$\underset{V(s)}{\arg\min}\,\mathbb{E}_{a\sim\pi_{\text{imp}}}[(Q(s, a) - V(s))^2]$$

*where $\pi_{\text{imp}}(a|s) \propto w(s, a)\mu(a|s)$, $w(s, a) = \frac{|f'(Q(s,a)-V^*(s))|}{|Q(s,a)-V^*(s)|}$.*

Different choices of loss function $f$ give rise to different implicit actor distributions. Recent variants of IQL like EQL (Garg et al., 2023) and SQL (Xu et al., 2023) matches the critic objectives for $f_\alpha(u) = \exp(u/\alpha) - u/\alpha$ and $f_\alpha(u) = \mathbb{1}(1 + u/2\alpha > 0)(1 + u/2\alpha) - u/\alpha$ respectively. Xu et al. (2023) pointed out that $\alpha$ in the loss functions can be seen as the regularization coefficient in the behavior-regularized MDP problem:

$$\max_\pi \mathbb{E}\left[\sum_{t=0}^\infty \gamma^t \left(r(s_t, a_t) - \alpha \cdot F(\frac{\pi(a_t|s_t)}{\mu(a_t|s_t)})\right)\right] \quad (5)$$

for some regularization function $F$.

In generalized IQL algorithms, the strength of regularization is usually controlled using a single scalar hyperparameter. In IQL (Kostrikov et al., 2022), computing a larger expectile $\tau \in (0, 1)$ can, in theory, better approximate the in-sample maximum $V \approx \max_{a'\in\mathcal{A},\, \mu(a'|s')>0} Q(s', a')$. This enables better filtering of suboptimal actions by using the advantage function $Q(s, a) - V(s)$ during policy extraction. Similarly, using a smaller $\alpha$ in Equation (5) (Xu et al., 2023) better approximates the unconstrained policy optimization problem.

## 4 VALUE LEARNING WITH IN-SAMPLE $V$-ENSEMBLE

### 4.1 OVERFITTING OF $V$-FUNCTION AND IN-SAMPLE ERROR ACCUMULATION

Intuitively, a higher $\tau$ (or a lower $\alpha$ in SQL (Xu et al., 2023)) should be used in mixed datasets in order to better approximate the in-sample maximum of $Q_\theta(s,a)$, thus filtering out suboptimal actions as much as possible. However, we show that the overfitting of $V$-function impedes the desired modification. Empirically, we find that the initial error of $Q_\theta(s,a)$ would be captured by an overfitted $V_\psi(s)$ and propagated among states through bootstrapping. The result is that in-sample offline RL algorithms remain susceptible to *catastrophic overestimation*, despite the complete avoidance of value queries for unseen actions. Also, the use of a higher $\tau$ (or a lower $\alpha$) makes $V_\psi(s)$ more sensitive to highly overestimated $Q_\theta(s,a)$ values, which exacerbated the overfitting of $V$-function.

To demonstrate the problem computationally, we train SQL agents on the D4RL (Fu et al., 2020) antmaze-umaze-d-v2 dataset for 1M updates. The result is shown in Figure 1. The AntMaze datasets typically require a larger $\tau$ (or a smaller $\alpha$) for effective value propagation. In vanilla SQL without additional regularization on the value function $V_\psi(s)$, taking $\alpha$ down to 0.1 results in numerical instabilities. To stabilize bootstrapping, the official implementation of SQL applied Dropout (Srivastava et al., 2014) with $p = 0.5$ and layer normalization (Ba et al., 2016) to $V_\psi(s)$. Without Dropout and layer normalization, the critic diverges with a small $\alpha = 0.1$ in SQL, and yet critic divergence could still happen after these modifications.

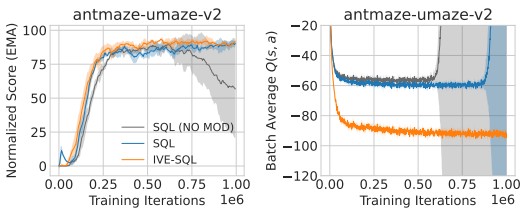

Figure 1: Normalized score and batch average $Q(s,a)$ of SQL and IVE-SQL. "SQL (no mod)" stands for SQL without value function Dropout and layer normalization.

*Remark* 4.1. We remark that the problem of $V$-function overfitting is related to the *advantage sparsity* defined as $s(Q, V; \mathcal{D}) = \mathbb{E}_{(s,a)\sim\mathcal{D}}[\mathbb{1}(Q(s,a) - V(s) \le 0)]$ (Xu et al., 2023). In regions with high epistemic uncertainty (Clements et al., 2019), $V(s)$ could easily overfit $Q(s,a)$, resulting in highly sparse advantage estimates. Note that for some variants like SQL (Xu et al., 2023) where the weight is simply $\max\{Q(s,a) - V(s), 0\}$, such state-action pair provides *zero* learning signal. In the limit, AWR is equivalent to behavior cloning when the advantage distribution is a Dirac delta $\delta(0)$ with mean zero, making it harder to learn selectively from the dataset.

### 4.2 VALUE LEARNING WITH IN-SAMPLE $V$-ENSEMBLE

To mitigate overfitting of the $V$-function, we propose using an ensemble of $m$ value functions $\{V_{\psi_i}(s)\}_{i=1}^m$ to perform in-sample updates. Given a generalized IQL loss function $f$, the $V$-functions are independently optimized using the following loss function:

$$\mathcal{L}_V^f(\psi_i) = \mathbb{E}_{(s,a)\sim\mathcal{D}}[f(Q_{\hat\theta}(s,a) - V_{\psi_i}(s))]. \tag{6}$$

The $Q$-function can then be learned by bootstrapping against an aggregate of the $V$-ensemble. To control $V$-function overfitting and stabilize bootstrapping, we can use the ensemble minimum:

$$\mathcal{L}_Q(\theta) = \mathbb{E}_{(s,a,s')\sim\mathcal{D}}[(r(s,a) + \gamma \min_i V_{\psi_i}(s') - Q_\theta(s,a))^2]. \tag{7}$$

To extract the policy from learned $Q_\theta(s,a)$ and $\{V_{\psi_i}(s)\}$, we can perform AWR (Peng et al., 2019) or its variants using the advantage estimate $\hat{A}(s,a) = Q_{\hat\theta}(s,a) - \mathbb{E}^c[V_{\psi_i}](s)$, where $\mathbb{E}^c[V_{\psi_i}]$ is the $c$-th quantile of $\{V_{\psi_i}\}_{i=1}^m$. AWR with $V$-ensemble maximizes the following objective:

$$\mathcal{L}_\pi^{\text{AWR}}(\phi) = \mathbb{E}_{(s,a)\sim\mathcal{D}}[\exp(\beta_{\text{AWR}}(Q_{\hat\theta}(s,a) - \mathbb{E}^c[V_{\psi_i}](s)))\log\pi_\phi(a|s)] \tag{8}$$

where $c \in [0,1]$ and $\beta_{\text{AWR}} \in [0,\infty]$ is an inverse temperature.

The effect of the $V$-ensemble on in-sample learning is twofold. First, we take the ensemble minimum during bootstrapping in Equation (7), which serves as a penalty for high-uncertainty states. The bootstrapping target can be interpreted as utilizing the lower-confidence bound (LCB) of $V$-predictions. Assume that $V(s)$ follows a Gaussian distribution with mean $\mu(s)$ and standard deviation $\sigma(s)$. The ensemble $\{V_i(s)\}_{i=1}^m$ contains samples of $V(s)$. Similar to (An et al., 2021), the expected ensemble minimum can be approximated as (Royston, 1982):

$$\mathbb{E}\left[\min_{1\leq i\leq m} V_i(s)\right]$$
$$\approx \mu(s) - \Phi^{-1}\left(\frac{m - \frac{\pi}{8}}{m - \frac{\pi}{4} + 1}\right)\sigma(s) \quad (9)$$

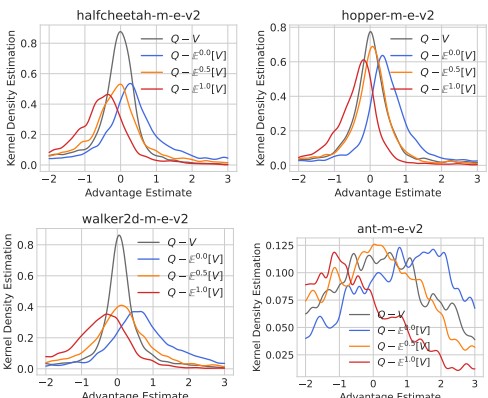

Figure 2: Kernel density estimation of advantage functions obtained by training on 4 datasets from the D4RL benchmark using single network SQL (gray) and $V$-ensemble (blue, red and orange). $\mathbb{E}^c[V]$ denotes the $c$-th quantile of the ensemble.

As we will show in our empirical evaluations, this penalty effectively mitigates the numerical instabilities encountered when a large $\tau$ (or small $\alpha$) is used. Second, the $V$-ensemble also reshapes the advantage distribution used by AWR. Unlike algorithms with out-of-distribution (OOD) evaluations (e.g. SAC-N (An et al., 2021)) in which uncertainty penalties are intuitive for OOD state-action pairs, the $V$-ensemble captures *in-sample epistemic uncertainty* which is less well understood. Here we provide an empirical study of the impact of $V$-ensemble on the advantage distribution. Figure 2 illustrates the distribution of advantage estimates obtained using Vanilla SQL and $V$-ensemble of size $m = 5$ on 4 D4RL Gym MuJoCo datasets, from which we can see that $V$-ensembles generally produce advantage distributions that are less sparse but sometimes *biased*. We find that the bias-sparsity trade-off controlled by $\mathbb{E}^c[\cdot]$ could significantly affect training outcome, and we will provide more empirical analyses in Section 7.

## 4.3 THEORETICAL ANALYSIS

To analyze the output dynamics of $Q$-functions with generalized IQL updates using the in-sample $V$-ensemble, we draw upon the work on the Neural Tangent Kernel (NTK) (Jacot et al., 2018; Lee et al., 2019; Ghasemipour & Gu, 2022). We will begin by establishing some additional assumptions and notations. First of all, Theorem 3.1 indicates that there exists an implicit policy $\pi_{\text{imp}}$ determined by $V^*$, such that the $V$-function can be equivalently learned by regressing towards $Q(s, \pi_{\text{imp}}(s))$ using a *MSE* loss. This leads to the following assumption:

**Assumption 4.2.** Given the current $Q$-network, the corresponding $\pi_{\text{imp}}$ can be sampled for any state $s$ appearing in the dataset $\mathcal{D} = \{(s, a, r, s')\}$. $\pi_{\text{imp}}(s)$ denotes a sample of $\pi_{\text{imp}}(a|s)$.

To facilitate our analysis, we also frame the problem as a fixed-policy evaluation featuring a slightly different learning target (justified by Equation (9)) by making the following assumptions.

**Assumption 4.3.** The implicit policy $\pi_{\text{imp}}$ is kept *fixed* throughout the value learning process.

**Assumption 4.4.** The regression target for $Q_\theta(s, a)$ is $r(s, a) + \gamma(\mathbb{E}_{\text{ens}}[V](s') - \sqrt{\mathbb{V}_{\text{ens}}[V](s')})$.

Next, we present notations relevant to our NTK-based analysis. Let $\mathcal{O}, \mathcal{O}', \mathcal{X}, R$ and $\tilde{\mathcal{X}}$ denote data matrices containing $s, s', (s, a), r$ and $(s, \pi_{\text{imp}}(s))$ where $(s, a, r, s')$ appears in the offline dataset $\mathcal{D}$. We consider NTK-parameterized $Q_\theta$ and $V_\psi$. For any two state-action data matrices $A$ and $B$, the *initial* NTK of $Q_\theta$ is given by $\hat{\Theta}_i^{(0)}(A, B) = \nabla_\theta Q_{\theta_i}(A) \cdot \nabla_\theta Q_{\theta_i}(B)^\top |_{t=0}$. At infinite width, the kernel converges to a deterministic one denoted by $\hat{\Theta}^{(0)}$. Analogously, we define the initial NTK (at infinite width) of $V_\psi$ acting on state data matrices as $\hat{\Theta}_V^{(0)}$. Let $C_{\text{imp}} = \hat{\Theta}^{(0)}(\tilde{\mathcal{X}}, \mathcal{X}) \cdot \hat{\Theta}^{(0)}(\mathcal{X}, \mathcal{X})^{-1}$ and $C_V = \hat{\Theta}_V^{(0)}(\mathcal{O}', \mathcal{O}) \cdot \hat{\Theta}_V^{(0)}(\mathcal{O}, \mathcal{O})^{-1}$. For policy evaluation, define $Q^{(0)} = Q_\theta$ and $V^{(0)} = V_{\psi_j}$ with parameters $\theta, \psi_j$ sampled from the initial weight distribution. Ensemble indices are specified if necessary. $\{Q^{(t)}\}$ and $\{V^{(t)}\}$ denote the sequence of intermediate estimates during training, where $V^{(t)}$ is updated using $Q^{(t)}, \forall t \geq 1$. Assume $\gamma \|C_V C_{\text{imp}}\| < 1$.

We are now ready to present our main theoretical result:

**Theorem 4.5.** *Let* $\Omega = C_V(Q^{(0)}(\tilde{\mathcal{X}}) - C_{imp}Q^{(0)}(\mathcal{X}))$, $B = \sqrt{\mathbb{E}_{ens}\left[\left(V^{(0)}(\mathcal{O}') - C_V \cdot V^{(0)}(\mathcal{O})\right)^2\right]}$.
*After* $t + 1$ *iterations of update (6) and (7) (with 4.4), the $Q$-estimate for* $(s, a) \sim \mathcal{X}$ *is given by:*

$$
\begin{aligned}
Q^{(t+1)}(\mathcal{X}) = &\left[1 + \cdots + (\gamma C_V C_{imp})^t\right] R \\
&+ \gamma \left[1 + \cdots + (\gamma C_V C_{imp})^{t-1}\right](\Omega - B) + O\left(\gamma^t \|C_V C_{imp}\|^t\right).
\end{aligned}
\tag{10}
$$

See Appendix A for a detailed proof. Note that $\tilde{\mathcal{X}}$ in $\Omega$ is likely to select high $Q^{(0)}$ values due to the implicit optimization of $\pi_{\text{imp}}$. And Theorem 4.5 indicates that the proposed $Q$-function update penalizes the "initial error" term $\Omega$ by $B \geq 0$, thus reducing the chance of catastrophic overestimation. Note that Assumption 4.3 is a strong one which is not satisfied in the actual IQL algorithm. The output dynamics of IQL under varying implicit policies remains an area of further study.

## 5 POLICY EXTRACTION WITH ADAPTIVE CLONING TEMPERATURE

Improving value learning alone can be helpful for the optimization of the implicit policy $\pi_{\text{imp}}$, but may not completely solve the over-regularization problem since AWR always imposes a KL-divergence constraint (Peng et al., 2019). Inspired by the automated entropy adjustment technique used in SAC (Haarnoja et al., 2018), we formulate a constrained optimization problem where the *average likelihood* of the state-action pairs from the dataset is constrained, while the likelihood at different states can vary. Formally, consider the following constrained optimization problem:

$$
\min_{\phi} \quad -\mathbb{E}_{s \sim \mathcal{D}, a \sim \pi_{\phi}}[Q_{\hat{\theta}}(s, a)] \quad \text{s.t.} \quad \mathbb{E}_{(s,a) \sim \mathcal{D}}[\log \pi_{\phi}(a|s)] \geq \mathcal{H}_{\mathcal{D}}
\tag{11}
$$

where $\mathcal{H}_{\mathcal{D}}$ is a given constant. The Lagrangian of Equation (11) is given by

$$
L(\phi, \beta) = -\mathbb{E}_{s \sim \mathcal{D}, a \sim \pi_{\phi}}[Q_{\hat{\theta}}(s, a)] + \beta(-\mathbb{E}_{(s,a) \sim \mathcal{D}}[\log \pi_{\phi}(a|s)] + \mathcal{H}_{\mathcal{D}}).
\tag{12}
$$

Similar to (Haarnoja et al., 2018), we perform approximate dual gradient descent on the constrained optimization problem, which alternates between optimizing the Lagrangian with respect to the primal variable $\pi$ and optimizing the dual variable $\beta$ by minimizing the following loss:

$$
\mathcal{L}(\beta) = \mathbb{E}_{(s,a) \sim \mathcal{D}}[\beta(\log \pi_{\phi}(a|s) - \mathcal{H}_{\mathcal{D}})].
\tag{13}
$$

Minimizing loss (13) increases the value of $\beta$ if $\log \pi_{\phi}(a|s) < \mathcal{H}_{\mathcal{D}}$ and vice versa. One potential problem of this approach is that if $\mathcal{H}_{\mathcal{D}}$ is overestimated, the behavior of (12) will gradually approach behavior cloning. We can alleviate this issue by combining (12) with weighted behavior cloning as:

$$
\mathcal{L}_{\pi}(\phi) = -\mathbb{E}_{s \sim \mathcal{D}, a \sim \pi_{\phi}}[Q_{\hat{\theta}}(s, a)] - \beta \mathbb{E}_{(s,a) \sim \mathcal{D}}[w(s, a) \log \pi_{\phi}(a|s)]
\tag{14}
$$

where $w(s, a)$ can be computed using learned $Q(s, a)$ and $V$-ensemble. As an example, for AWR:

$$
w(s, a) = \exp(Q_{\hat{\theta}}(s, a) - \mathbb{E}^c[V_{\psi_i}](s)).
\tag{15}
$$

By optimizing the loss in Equation (14), in the worst case where $\beta$ blows up due to an overestimated $\mathcal{H}_{\mathcal{D}}$, we can still get similar behavior to weighted behavior cloning.

*Remark* 5.1. Estimating the optimal $\mathcal{H}_{\mathcal{D}}$ is difficult, since the average log-likelihood over $\mathcal{D}$ of the best policy $\pi_{\mathcal{D}}^*$ that can be learnt from $\mathcal{D}$ varies greatly with the quality of $\mathcal{D}$. We describe the tuning procedure for $\mathcal{H}_{\mathcal{D}}$ in Appendix B.

## 6 ALGORITHM SUMMARY

We now describe our complete algorithm which combines our modifications on value learning and policy extraction. We refer to the resulting algorithm as Actor-Critic with Temperature adjustment and In-sample Value Ensemble (ACTIVE) and summarize the approach in Algorithm 1.

---

**Algorithm 1** ACTIVE

**Hyperparameters:** $f = f_{\alpha}$, $m$, $\mathcal{H}_{\mathcal{D}}$, LR $\lambda$, $\lambda_{\beta}$, EMA $\eta$.
**Initialize:** $\phi, \theta, \hat{\theta}, \{\psi_i\}_{i=1}^m, \beta, \mathcal{D}$.
**for** each gradient step **do**
    $\psi_i \leftarrow \psi_i - \lambda \nabla_{\psi_i} \mathcal{L}_V^f(\psi_i)$ (Equation (6))
    $\theta \leftarrow \theta - \lambda \nabla_{\theta} \mathcal{L}_Q(\theta)$ (Equation (7))
    $\hat{\theta} \leftarrow (1 - \eta)\hat{\theta} + \eta\theta$
    $\beta \leftarrow \beta - \lambda_{\beta} \nabla_{\beta} \beta$ (Equation (13))
    $\phi \leftarrow \phi - \lambda \nabla_{\phi} \mathcal{L}_{\pi}(\phi)$ (Equation (14))
**end for**

---

Table 2: Average normalized score over the final 10 evaluations and 5 seeds. $\pm$ captures the standard deviation over seeds. The results of CQL, TD3+BC, IQL and SQL are taken from the authors. The "-I" and "-S" suffixes indicate the use of IQL-style and SQL-style $V$-function losses, respectively. The highest scores are **bolded**. Additionally, scores higher than the corresponding reproduced baseline (IQL for ACTIVE-I, SQL for ACTIVE-S) are marked in blue.

| Dataset | CQL | TD3+BC | IQL | SQL | IQL (Reproduced) | SQL (Reproduced) | ACTIVE-I | ACTIVE-S |
|---|---|---|---|---|---|---|---|---|
| halfcheetah-m | 44.0 ±0.8 | 48.3 ±0.3 | 47.4 ±0.2 | 48.3 ±0.2 | 47.4 ±0.1 | 47.8 ±0.0 | 50.6 ±0.1 | **52.3 ±0.2** |
| hopper-m | 58.5 ±2.1 | 59.3 ±4.2 | 66.2 ±5.7 | 75.5 ±3.4 | 64.4 ±2.7 | 63.8 ±2.3 | 80.1 ±3.8 | **86.1 ±3.4** |
| walker2d-m | 72.5 ±0.8 | 83.7 ±2.1 | 78.3 ±8.7 | 84.2 ±4.6 | 80.1 ±1.3 | 82.8 ±0.4 | 84.8 ±0.7 | **87.2 ±1.0** |
| halfcheetah-m-r | 45.5 ±0.5 | 44.6 ±0.5 | 44.2 ±1.2 | 44.8 ±0.7 | 43.6 ±0.2 | 43.9 ±0.3 | 51.1 ±0.3 | **51.7 ±0.2** |
| hopper-m-r | 95.0 ±6.4 | 60.9 ±18.8 | 94.7 ±8.6 | 101.7 ±3.3 | 86.0 ±13.4 | 85.6 ±12.3 | 101.1 ±1.6 | **102.8±0.7** |
| walker2d-m-r | 77.2 ±5.5 | 81.8 ±5.5 | 73.8 ±7.1 | 77.2 ±3.8 | 70.3 ±5.6 | 69.2 ±6.3 | **85.4 ±3.5** | 79.3 ±4.6 |
| halfcheetah-m-e | 90.7 ±4.3 | 90.7 ±4.3 | 86.7 ±5.3 | **94.0 ±0.4** | 88.2 ±1.6 | 91.2 ±1.2 | 93.2 ±0.4 | 92.9 ±1.0 |
| hopper-m-e | 105.4 ±6.8 | 98.0 ±9.4 | 91.5 ±14.3 | **111.8 ±2.2** | 98.7 ±7.9 | 104.9 ±5.4 | 89.2 ±10.9 | 109.9 ±2.0 |
| walker2d-m-e | 109.6 ±0.7 | 110.1 ±0.5 | 109.6 ±1.0 | 110.0 ±0.8 | 110.0 ±0.4 | 111.2 ±0.1 | **112.0 ±0.4** | 111.7 ±0.3 |
| antmaze-u | 74.0 | 78.6 | 87.5 ±2.6 | **92.2 ±1.4** | 85.8 ±3.2 | 90.0 ±3.1 | 91.6 ±1.4 | 90.8 ±1.5 |
| antmaze-u-d | **84.0** | 71.4 | 62.2 ±13.8 | 74.0 ±2.3 | 60.6 ±14.2 | 49.6 ±6.9 | 78.4 ±5.1 | 43.8 ±22.1 |
| antmaze-m-p | 61.2 | 10.6 | 71.2 ±7.3 | **80.2 ±3.7** | 76.2 ±6.2 | 72.8 ±3.2 | 74.6 ±5.5 | 78.8 ±1.7 |
| antmaze-m-d | 53.7 | 3.0 | 70.0 ±10.9 | 75.1 ±4.2 | 73.8 ±3.7 | 62.2 ±8.1 | **75.2 ±2.2** | 67.0 ±3.3 |
| antmaze-l-p | 15.8 | 0.2 | 39.6 ±5.8 | 50.2 ±4.8 | 48.2 ±6.5 | 38.2 ±5.2 | 51.2 ±5.8 | **51.2 ±5.7** |
| antmaze-l-d | 14.9 | 0.0 | 47.5 ±9.5 | **52.3 ±5.2** | 47.2 ±3.5 | 41.6 ±4.4 | 48.2 ±5.8 | 42.6 ±3.2 |
| kitchen-c | 43.8 ±11.2 | - | 61.4 ±9.5 | **76.4 ±8.7** | 67.0 ±3.4 | 60.4 ±1.3 | 68.2 ±3.6 | 66.5 ±3.3 |
| kitchen-p | 49.8 ±10.1 | - | 46.1 ±8.5 | 72.5 ±7.4 | 58.8 ±8.7 | 70.8 ±4.1 | 69.6 ±3.0 | **73.5 ±3.0** |
| kitchen-m | 51.0 ±6.5 | - | 52.8 ±4.5 | 67.4 ±5.4 | 47.9 ±3.7 | 46.9 ±11.8 | 52.1 ±2.9 | **73.2 ±1.4** |

## 7 EXPERIMENTS

In this section, we present empirical evaluations of ACTIVE against baseline algorithms. We first compare our method with baseline model-free offline RL methods on the D4RL benchmark. We then analyze the effect of in-sample $V$-ensemble (IVE) and adaptive cloning temperature (ACT) in ablation studies. Finally, we present an empirical comparison of the running time of ACTIVE and prior in-sample learning methods.

### 7.1 COMPARISONS ON OFFLINE RL BENCHMARKS

**Comparisons and Baselines.** First, we evaluate our method on Gym MuJoCo, AntMaze and Kitchen datasets of the D4RL benchmark (Fu et al., 2020). For a fair comparison, we follow the setting of IQL (Kostrikov et al., 2022) in which Gym MuJoCo and AntMaze datasets take the "-v2" version while Kitchen datasets take the "-v0" version. Note that AntMaze and Kitchen datasets contain fewer near-optimal trajectories than Gym MuJoCo datasets, and learning ef-

Table 1: Comparison against ensemble-based algorithms with similar ensemble sizes ($m$). The results of RORL and MSG are taken from the authors.

| Dataset | RORL ($m = 10$) | MSG ($m = 4$) | ACTIVE-I ($m \leq 7$) | ACTIVE-S ($m \leq 7$) |
|---|---|---|---|---|
| antmaze-u | 96.7 ± 1.9 | 98.6 ± 1.4 | 91.6 ± 1.4 | 90.8 ± 1.5 |
| antmaze-u-d | 90.7 ± 2.9 | 76.6 ± 7.6 | 78.4 ± 5.1 | 43.8 ± 22.1 |
| antmaze-m-p | 76.3 ± 2.5 | 83.0 ± 7.1 | 74.6 ± 5.5 | 78.8 ± 1.7 |
| antmaze-m-d | 69.3 ± 3.3 | 83.0 ± 6.2 | 75.2 ± 2.2 | 67.0 ± 3.3 |
| antmaze-l-p | 16.3 ± 11.1 | 46.8 ± 14.7 | 51.2 ± 5.8 | 51.2 ± 5.7 |
| antmaze-l-d | 41.0 ± 10.7 | 58.2 ± 9.6 | 48.2 ± 5.8 | 42.6 ± 3.2 |
| Average | 65.1 | 74.4 | 69.9 | 62.3 |

fective policies from them requires "stitching" together sub-trajectories. We compare our method with popular model-free offline RL methods, including CQL (Kumar et al., 2020), TD3+BC (Fujimoto & Gu, 2021), IQL (Kostrikov et al., 2022) and SQL (Xu et al., 2023). The results are presented in Table 2. We can see that our method performs comparably to the best-performing prior method and outperforms the corresponding in-sample baselines (IQL for ACTIVE-I, SQL for ACTIVE-S), especially on suboptimal or diverse datasets (e.g. the m/m-r datasets, antmaze-u-d, kitchen-m).

Additionally, we compare ACTIVE with popular ensemble-based algorithms, including RORL (Yang et al., 2022) and MSG (Ghasemipour & Gu, 2022) on AntMaze tasks, and SAC-N (An et al., 2021) on Kitchen tasks. While it is known that ensemble-based methods like SAC-N performs well on MuJoCo tasks, AntMaze and Kitchen tasks can be more realistic and challenging (Fu et al., 2020). The results on AntMaze with similar ensemble sizes are shown in Table 1. Note that MSG does not rely solely on ensembles but instead requires a CQL-like regularizer. In contrast, ACTIVE does not require additional explicit regularization. Due to the lack of official results on Kitchen tasks, we run SAC-N with ensemble size $m = 20$ and 40. In both cases we find that SAC-N rapidly diverges and failed to learn any meaningful policy, as shown in Table 8. Note that an ensemble of size $m = 40$ is already much larger than the value ensemble required in our method (typically $5 \leq m \leq 7$).

**Effect of In-sample $V$-Ensemble (IVE).** We demonstrate the effect of $V$-ensemble and its interaction with $\tau$ (or $\alpha$). First we compare IVE-S and SQL on the simpler antmaze-u dataset. The result is shown in Figure 1. Note that Dropout can also be used to capture epistemic uncertainty (Gal & Ghahramani, 2016) but in SQL it is only used to mitigate overfitting. In ACTIVE we take a different approach of taking the ensemble minimum in bootstrapping and policy extraction. It is evident that our method successfully prevents catastrophic errors and is more consistent in reaching higher scores. We present additional results for reducing catastrophic overestimation on the antmaze-u-d dataset shown in Figure 11 in Appendix C. For harder tasks, such as "diverse" versions of the datasets, a larger $\tau$ is desired in IQL to extract better policies but performs poorly due to numerical instabilities. In ACTIVE, a fixed-size $V$-ensemble $m = 6$ allows for a larger expectile $\tau \in \{0.9, 0.95, 0.99\}$, while a larger ensemble size $m \in \{1, 2, 3, 6\}$ stabilizes bootstrapping off large expectile estimates $\tau = 0.99$, as shown in Figure 3. This suggests that on datasets that require more value propagation, performance gains can be obtained by using a $V$-ensemble and gradually increase $\tau$.

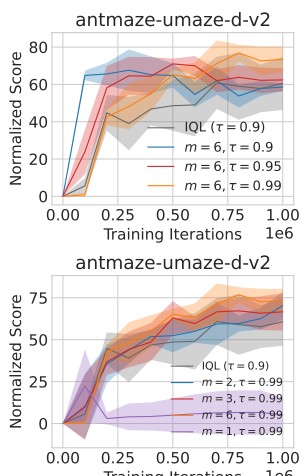

Figure 3: Normalized score of IVE-IQL with different ensemble size $m$ and expectile $\tau$.

In Section 4.2, we mentioned that IVE introduces a bias-sparsity trade-off controlled by $c$ in $\mathbb{E}^c[V_i]$. While the advantage bias could be optimistic when $c < 0.5$, The value functions learned by IVE is still *pessimistic* due to the target $r + \gamma \min_i V_i$. Here we demonstrate the impact of $\mathbb{E}^c[V_i]$ on policy performance. As shown in Figure 4, $\mathbb{E}^c[V_i]$ could effectively aid training by trading bias off for sparsity control. For most MuJoCo datasets we find that a large $c = 1.0$ is beneficial for action filtering. In cases where the advantage is more sparse (e.g. antmaze-u-d), a small $c$ could facilitate learning. To promote understanding of the in-sample epistemic uncertainty, we present additional visualizations shown in Figure 15 in Appendix C.

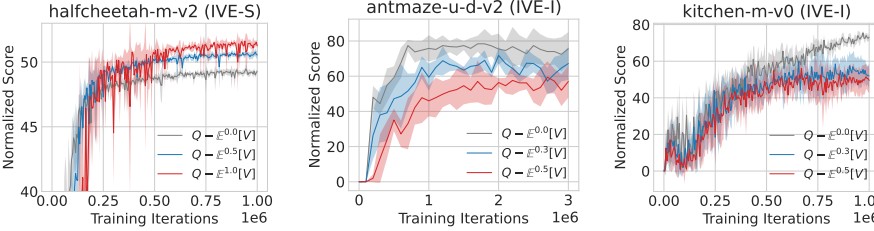

Figure 4: Normalized score of IVE-I/S with advantage estimates computed using different ensemble aggregates. $\mathbb{E}^c[V]$ denotes the $c$-th quantile of the $V$-ensemble.

**Effect of Adaptive Cloning Temperature (ACT).** As discussed in Section 5, over-regularization can happen when the dataset is skewed towards suboptimal policies, and the average likelihood of dataset actions given by the learned policy can overshoot. We present learning curves of SQL and ACT-S on suboptimal datasets in Figure 5. During training, ACT successfully adjusts batch average $\log \pi(a|s)$ to a neighborhood of the target likelihood $\mathcal{H}_{\mathcal{D}}$. We can see that in the early stages of training, ACT performs similarly to weighted behavior cloning. On the other hand, a slightly lower $\mathcal{H}_{\mathcal{D}}$ allows for more generalization and results in better performance.

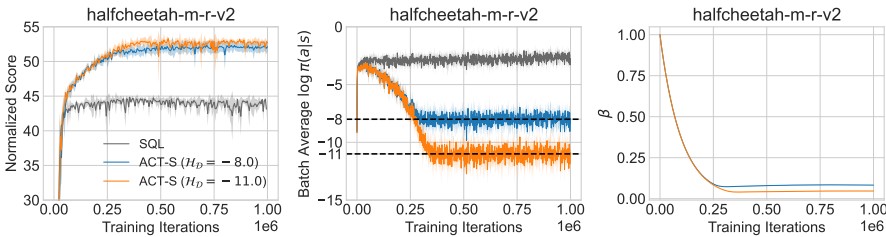

Figure 5: Comparison between SQL and ACT-S with varied $\mathcal{H}_{\mathcal{D}}$ on a suboptimal dataset.

## 7.2 ABLATION STUDIES

In this section, we present the results of an ablation study over the components of ACTIVE, including In-sample $V$-Ensemble (IVE) and Adaptive Cloning Temperature (ACT). Learning curves and performance profiles are shown in Figures 6(a) and 6(b). We can see that the performance improvement on compositional datasets like AntMaze is mainly attributed to IVE, while ACT mainly improves performance on suboptimal datasets in the MuJoCo suite. We present additional results on Kitchen tasks in Figure 12 in Appendix C.

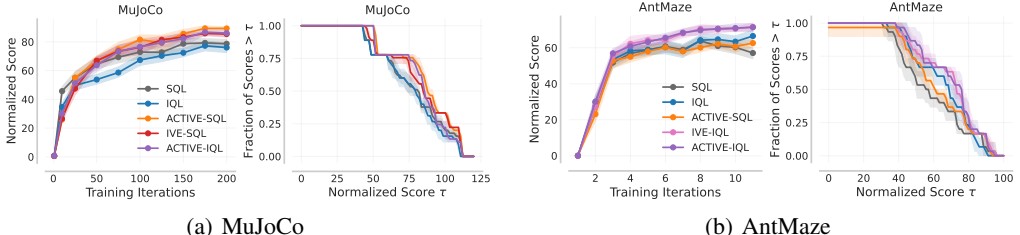

(a) MuJoCo                                (b) AntMaze

Figure 6: Average learning curves with stratified bootstrap confidence intervals (CIs) and performance profiles generated using rliable (Agarwal et al., 2021) on (a) MuJoCo and (b) AntMaze tasks.

## 7.3 RUN TIME

In this section, we compare the run time of ACTIVE and baseline in-sample algorithms by running them on the same hardware and software configuration for 1M steps. The result is displayed in Table 3. We can see that while ACTIVE introduces ensembles for uncertainty estimation, for small networks the run time does not increase significantly thanks to parallelization.

Table 3: Run time of ACTIVE and baselines on the halfcheetah-medium-v2 dataset.

| Algorithm | Iter/s | Time (min) |
|---|---|---|
| IQL | 744 | 22.27 |
| SQL | 727 | 22.79 |
| ACTIVE-I | 650 | 25.89 |
| ACTIVE-S | 627 | 26.48 |

## 8 DISCUSSION

Although we have shown in Section 7.1 that adaptive temperature adjustment does improve final performance on certain datasets, a SAC-style policy extraction loss in Equation (14) with an in-sample critic does not achieve similar results as EDAC (An et al., 2021) on MuJoCo tasks. We argue that critics trained in an in-sample manner may only generalize in a close neighborhood of dataset samples. And for effective generalization to take place in offline RL tasks, it may be necessary to perform out-of-sample updates. Given that in-sample critics can be empirically more accurate (Fu et al., 2022), how to combine in-sample learning and out-of-sample learning for balancing generalization and regularization may be worth investigating. These discussions are beyond the scope of this work, and we leave them for future works.

Recent generalizations of IQL such as EQL (Garg et al., 2023) studied the Bellman error distribution in MDPs and proposed using Gumbel regression to learn the $V$-function. And in ACTIVE, the Bellman error distribution could be affected by the $V$-ensemble. We analyze the Bellman error distribution of single-net IQL and IVE in Figure 13 in Appendix C, in which we do not observe significant difference between the two set of distributions. Nevertheless, the Bellman error distribution in offline RL methods may be worth investigating and remains an area of further study.

## 9 CONCLUSION

We have introduced ACTIVE, an offline RL algorithm designed to alleviate over-regularization in existing in-sample learning methods. ACTIVE provides an ensemble-based value learning scheme and a policy extraction loss based on dual gradient descent that can be applied to any variant of implicit TD backups. Experimental results show that our method effectively mitigates overfitting of $V$-function and prevents over-regularization on suboptimal datasets. Compared to existing in-sample offline RL algorithms, ACTIVE exhibits better stability and final policy performance on a variety of offline datasets.

ACKNOWLEDGEMENTS

This work was supported by the National Science and Technology Major Project under Grant 2022ZD0116401, the National Natural Science Foundation of China under Grant 62141605, and the Fundamental Research Funds for the Central Universities, China. We thank all the anonymous reviewers for their insightful comments.

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

## A    PROOF OF THEOREM 4.5

Following the assumptions and notations introduced in Section 4.3, we present the proof of Theorem 4.5. The proof builds on existing work on NTK and ensemble-based offline RL methods (Lee et al., 2019; Ghasemipour & Gu, 2022).

*Proof.* Consider the linearization (w.r.t. parameters) of $Q^{(0)}$ at initialization:

$$Q_{\text{lin}}(s, a) := Q^{(0)}(s, a) + \nabla_\theta Q^{(0)}(s, a) \cdot \left(\theta_{\text{lin}} - \theta^{(0)}\right). \tag{16}$$

Let $Q_{\text{lin}}^{(0)} = Q_{\text{lin}}$, and consider performing updates on the linearized network to get $\{Q_{\text{lin}}^{(t)}\}$. Lee et al. (2019) shows that with a MSE loss and infinite network width, subject to technical conditions on the learning rate, the predictions of the trained networks $Q^{(t)}$ coincide with the predictions of the trained linearized networks $Q_{\text{lin}}^{(t)}$, i.e. $\forall(s, a), t, Q_{\text{lin}}^{(t)}(s, a) = Q^{(t)}(s, a)$. Since we have augmented $V$ to train on $\tilde{\mathcal{X}}$ using a MSE loss by Theorem 3.1 and Assumption 4.2, we can analogously define the linearized $V_{\text{lin}}^{(0)}$ and analyze the dynamics of $V_{\text{lin}}^{(t)}$ instead:

$$V_{\text{lin}}(s) := V^{(0)}(s) + \nabla_\psi V^{(0)}(s) \cdot \left(\psi_{\text{lin}} - \psi^{(0)}\right). \tag{17}$$

**Learning $V$.** Following Lee et al. (2019) (Section 2.2, Equations 9-11, by performing continuous-time gradient descent to the limit of $t \to \infty$ so that $(I - e^{-\eta\hat{\Theta}_0 t}) \to I$), we can derive:

$$\begin{aligned} V_{\text{lin}}^{(t)}(\mathcal{O}') &= V^{(0)}(\mathcal{O}') + \hat{\Theta}_V^{(0)}(\mathcal{O}', \mathcal{O}) \cdot \hat{\Theta}_V^{(0)}(\mathcal{O}, \mathcal{O})^{-1} \cdot \left(Q_{\text{lin}}^{(t)}(\tilde{\mathcal{X}}) - V^{(0)}(\mathcal{O})\right) \\ &= V^{(0)}(\mathcal{O}') + C_V \cdot \left(Q_{\text{lin}}^{(t)}(\tilde{\mathcal{X}}) - V^{(0)}(\mathcal{O})\right). \end{aligned} \tag{18}$$

Under the NTK setting, $\forall s, \mathbb{E}_{\text{ens}}[V^{(0)}(s)] = 0$. Therefore we have:

$$\mathbb{E}_{\text{ens}}[V_{\text{lin}}^{(t)}(\mathcal{O}')] = \mathbb{E}_{\text{ens}}\left[V^{(0)}(\mathcal{O}') + C_V \cdot \left(Q_{\text{lin}}^{(t)}(\tilde{\mathcal{X}}) - V^{(0)}(\mathcal{O})\right)\right] = C_V \cdot Q_{\text{lin}}^{(t)}(\tilde{\mathcal{X}})$$

$$\begin{aligned} \mathbb{V}_{\text{ens}}[V_{\text{lin}}^{(t)}(\mathcal{O}')] &= \mathbb{E}_{\text{ens}}\left[\left(V^{(0)}(\mathcal{O}') + C_V \cdot \left(Q_{\text{lin}}^{(t)}(\tilde{\mathcal{X}}) - V^{(0)}(\mathcal{O})\right)\right)^2\right] - (C_V \cdot Q_{\text{lin}}^{(t)}(\tilde{\mathcal{X}}))^2 \\ &= \mathbb{E}_{\text{ens}}\left[\left(V^{(0)}(\mathcal{O}') - C_V \cdot V^{(0)}(\mathcal{O})\right)^2\right]. \end{aligned} \tag{19}$$

To simplify the notation, let $B := \sqrt{\mathbb{V}_{\text{ens}}[V_{\text{lin}}^{(t)}(\mathcal{O}')]}$.

**Bootstrapping $Q$ against $V$.** By Assumption 4.4, the data matrix containing the learning target for $Q$ is $\mathcal{Y}^{(t)} = R + \gamma \cdot \text{LCB}(V_{\text{lin}}^{(t)}(\mathcal{O}'))$, where

$$\forall s', \text{LCB}(V_{\text{lin}}^{(t)}(s')) = \mathbb{E}_{\text{ens}}[V_{\text{lin}}^{(t)}(s')] - \sqrt{\mathbb{V}_{\text{ens}}[V_{\text{lin}}^{(t)}(s')]}. \tag{20}$$

Following Lee et al. (2019) (Section 2.2, Equation 9, $t \to \infty$), we can derive:

$$\begin{aligned} Q_{\text{lin}}^{(t+1)}(\mathcal{X}) &= \mathcal{Y}^{(t)} \\ Q_{\text{lin}}^{(t+1)}(\tilde{\mathcal{X}}) &= Q^{(0)}(\tilde{\mathcal{X}}) + \hat{\Theta}^{(0)}(\tilde{\mathcal{X}}, \mathcal{X}) \cdot \hat{\Theta}^{(0)}(\mathcal{X}, \mathcal{X})^{-1} \cdot \left(\mathcal{Y}^{(t)} - Q^{(0)}(\mathcal{X})\right) \\ &= Q^{(0)}(\tilde{\mathcal{X}}) + C_{\text{imp}} \cdot \left(\mathcal{Y}^{(t)} - Q^{(0)}(\mathcal{X})\right). \end{aligned} \tag{21}$$

Analogous to the proof of Theorem 3.1 in Ghasemipour & Gu (2022), we can recursively compute $\mathcal{Y}^{(t)}$:

$$\mathcal{Y}^{(t)} = R + \gamma \cdot \text{LCB}(V_{\text{lin}}^{(t)}(\mathcal{O}'))$$

$$= R + \gamma \left[ \mathbb{E}_{\text{ens}}[V_{\text{lin}}^{(t)}(\mathcal{O}')] - \sqrt{\mathbb{V}_{\text{ens}}[V_{\text{lin}}^{(t)}(\mathcal{O}')]} \right]$$

$$= R + \gamma \left[ C_V \cdot Q_{\text{lin}}^{(t)}(\tilde{\mathcal{X}}) - B \right]$$

$$= R + \gamma \left[ C_V \cdot \left( Q^{(0)}(\tilde{\mathcal{X}}) + C_{\text{imp}} \cdot \left( \mathcal{Y}^{(t-1)} - Q^{(0)}(\mathcal{X}) \right) \right) - B \right]$$

$$= R + \gamma(\Omega - B) + \gamma C_V C_{\text{imp}} \mathcal{Y}^{(t-1)}$$

$$= \cdots$$

$$= \left[ 1 + \cdots + (\gamma C_V C_{\text{imp}})^{t-1} \right] (R + \gamma(\Omega - B)) + (\gamma C_V C_{\text{imp}})^t \left( R + \gamma \text{LCB}(V_{\text{lin}}^{(0)}(\mathcal{O}')) \right)$$

$$= \left[ 1 + \cdots + (\gamma C_V C_{\text{imp}})^t \right] R + \gamma \left[ 1 + \cdots + (\gamma C_V C_{\text{imp}})^{t-1} \right] (\Omega - B) + O\left( \gamma^t \|C_V C_{\text{imp}}\|^t \right). \tag{22}$$

Plugging into Equation (21) and using the NTK property, we have:

$$Q^{(t+1)}(\mathcal{X}) = Q_{\text{lin}}^{(t+1)}(\mathcal{X}) = \left[ 1 + \cdots + (\gamma C_V C_{\text{imp}})^t \right] R + O\left( \gamma^t \|C_V C_{\text{imp}}\|^t \right)$$
$$+ \gamma \left[ 1 + \cdots + (\gamma C_V C_{\text{imp}})^{t-1} \right] (\Omega - B). \tag{23}$$

$\square$

# B    IMPLEMENTATION DETAILS

**General.** We implement ACTIVE and reproduce IQL (Kostrikov et al., 2022) and SQL (Xu et al., 2023) based on the author-provided source code. We mainly tune the implicit regularization level ($\alpha$ or $\tau$) along with ensemble size $m$ and target likelihood $\mathcal{H}_\mathcal{D}$ while most remaining hyperparameters remained unchanged from the corresponding baseline (IQL for ACTIVE-I, SQL for ACTIVE-S). For learning curves presented in the paper, we evaluate the agent for 10 episodes every 5000 steps. For benchmark results in Table 2, we follow IQL and SQL by averaging over 10 evaluations every 5000 training steps on MuJoCo/Kitchen, and averaging over 100 evaluations every 0.1M training steps on AntMaze. The full hyperparameters setting is shown in Tables 4 to 7.

**Ensemble Aggregate for Bootstrapping.** We use $\min_i V_{\psi_i}(s)$ as ensemble aggregate for bootstrapping on most D4RL datasets. On walker2d-medium-replay-v2, we use $\frac{1}{m} \sum_i^m V_{\psi_i}(s)$ instead as it empirically performs better. The full setup can be found in Table 7.

**Importance Weight in ACTIVE.** To isolate the effect of weight distribution and weight scaling (which is controlled by adaptive temperature $\beta$), for every batch we normalize $w(s, a)$ in Equation (14) by dividing each element by the batch mean (mean + $\epsilon$, $\epsilon = 1 \times 10^{-4}$ for ACTIVE-S).

**Tuning $\mathcal{H}_\mathcal{D}$.** In this work, we tune $\mathcal{H}_\mathcal{D}$ for each different dataset. To choose $\mathcal{H}_\mathcal{D}$ in an offline manner, we first run IVE-SQL/IQL to obtain the average log likelihood $\mathcal{H}_0$ evaluated on the dataset (estimated using the final minibatches). And then starting with $\mathcal{H}_\mathcal{D} = \mathcal{H}_0$ we decrease $\mathcal{H}_\mathcal{D}$ until the turning point of $\beta$ (i.e. when $\beta$ starts to decrease) gets lower than $1/2$ of the total training iterations.

**Software.** We use the following software versions:

- D4RL 1.1 (Fu et al., 2020) (Apache-2.0 license)
- Jax 0.4.9 (Bradbury et al., 2018) (Apache-2.0 license)
- MuJoCo 2.1.0 (Todorov et al., 2012) (Apache-2.0 license)
- Gym 0.23.1 (Brockman et al., 2016) (MIT license)

**Hardware.** We use the following hardware:

- NVIDIA RTX 3090
- Intel(R) Xeon(R) Silver 4216 CPU @ 2.10GHz

Table 4: ACTIVE-I/S general hyperparameters.

| Hyperparameter | Value |
|---|---|
| Actor learning rate | $3 \times 10^{-4}$ 
 $2 \times 10^{-4}$ for AntMaze in ACTIVE-S |
| Critic learning rate | $3 \times 10^{-4}$ 
 $2 \times 10^{-4}$ for AntMaze in ACTIVE-S |
| Value learning rate | $3 \times 10^{-4}$ 
 $2 \times 10^{-4}$ for AntMaze in ACTIVE-S |
| Batch size | 256 |
| Optimizer | Adam |
| Network (all) | 3 layers ReLU activated MLPs with 256 units |
| Discount $\gamma$ | 0.99 |
| Polyak $\eta$ | 0.005 |
| Layer normalization | Off |
| Value Dropout | Off |
| Actor Dropout | Off ($p = 0.1$ for Kitchen) |

Table 5: ACTIVE hyperparameters used in Kitchen domain.

| Dataset | ACTIVE-I | | | | | ACTIVE-S | | | |
|---|---|---|---|---|---|---|---|---|---|
| | $\tau$ | $\beta_{\text{AWR}}$ | $m$ | $c$ | $\mathcal{H}_{\mathcal{D}}$ | $\alpha$ | $m$ | $c$ | $\mathcal{H}_{\mathcal{D}}$ |
| kitchen-complete-v0 | 0.97 | 0.3 | 5 | 0.0 | Off | 0.05 | 5 | 0.0 | Off |
| kitchen-partial-v0 | 0.9 | 0.5 | 5 | 0.0 | Off | 0.3 | 5 | 0.0 | Off |
| kitchen-mixed-v0 | 0.95 | 0.3 | 5 | 0.0 | Off | 0.1 | 7 | 0.0 | Off |

Table 6: ACTIVE hyperparameters used in AntMaze domain.

| Dataset | ACTIVE-I | | | | | ACTIVE-S | | | |
|---|---|---|---|---|---|---|---|---|---|
| | $\tau$ | $\beta_{\text{AWR}}$ | $m$ | $c$ | $\mathcal{H}_{\mathcal{D}}$ | $\alpha$ | $m$ | $c$ | $\mathcal{H}_{\mathcal{D}}$ |
| antmaze-umaze-v2 | 0.99 | 3.0 | 5 | 0.0 | -5.25 | 0.02 | 5 | 0.0 | -5.25 |
| antmaze-umaze-diverse-v2 | 0.99 | 10.0 | 6 | 0.0 | -4.0 | 0.5 | 7 | 0.0 | -4.0 |
| antmaze-medium-play-v2 | 0.99 | 10.0 | 6 | 0.0 | -5.0 | 0.01 | 5 | 0.0 | -5.0 |
| antmaze-medium-diverse-v2 | 0.99 | 10.0 | 5 | 0.0 | -5.0 | 0.02 | 5 | 0.0 | -5.0 |
| antmaze-large-play-v2 | 0.99 | 10.0 | 6 | 0.0 | -5.25 | 0.02 | 5 | 0.0 | -5.25 |
| antmaze-large-diverse-v2 | 0.99 | 10.0 | 6 | 0.0 | -5.3 | 0.01 | 5 | 0.0 | -5.3 |

Table 7: ACTIVE hyperparameters used in MuJoCo domain.

| Dataset | ACTIVE-I | | | | | ACTIVE-S | | | | Ens. agg. for $Q$-update |
|---|---|---|---|---|---|---|---|---|---|---|
| | $\tau$ | $\beta_{\text{AWR}}$ | $m$ | $c$ | $\mathcal{H}_{\mathcal{D}}$ | $\alpha$ | $m$ | $c$ | $\mathcal{H}_{\mathcal{D}}$ | |
| halfcheetah-medium-v2 | 0.9 | 3.0 | 5 | 1.0 | 1.0 | 0.03 | 5 | 1.0 | -1.0 | min |
| hopper-medium-v2 | 0.9 | 3.0 | 5 | 1.0 | 0.0 | 0.1 | 5 | 1.0 | -0.9 | min |
| walker2d-medium-v2 | 0.9 | 3.0 | 5 | 1.0 | 1.0 | 0.1 | 5 | 1.0 | 0.0 | min |
| halfcheetah-medium-replay-v2 | 0.9 | 3.0 | 5 | 1.0 | -11.0 | 0.1 | 5 | 1.0 | -11.0 | min |
| hopper-medium-replay-v2 | 0.9 | 3.0 | 5 | 1.0 | -5.0 | 0.1 | 5 | 1.0 | -5.0 | min |
| walker2d-medium-replay-v2 | 0.9 | 3.0 | 5 | 1.0 | -7.0 | 1.0 | 5 | 0.0 | -3.0 | mean |
| halfcheetah-medium-expert-v2 | 0.7 | 3.0 | 5 | 0.0 | 1.75 | 1.0 | 5 | 0.0 | 1.75 | min |
| hopper-medium-expert-v2 | 0.7 | 3.0 | 5 | 0.0 | 0.0 | 1.0 | 5 | 0.0 | 0.0 | min |
| walker2d-medium-expert-v2 | 0.7 | 3.0 | 5 | 0.0 | 1.75 | 1.0 | 5 | 0.0 | 1.75 | min |

# C   MORE RESULTS

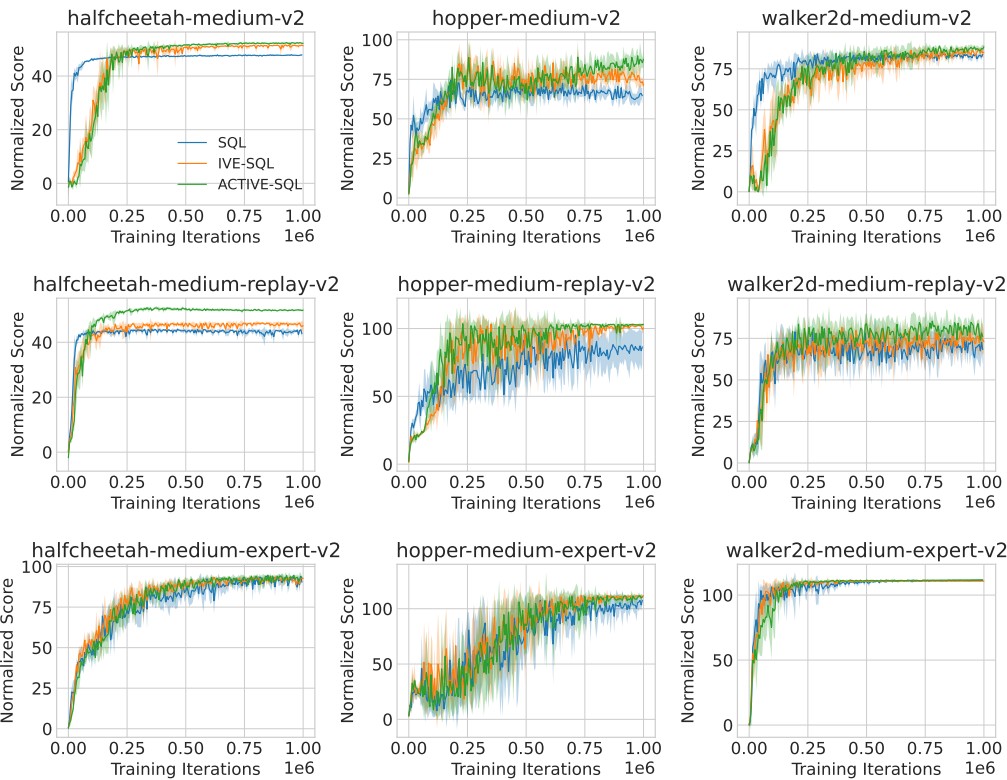

Figure 7: Learning curves of ACTIVE-S, IVE-S and SQL on MuJoCo datasets.

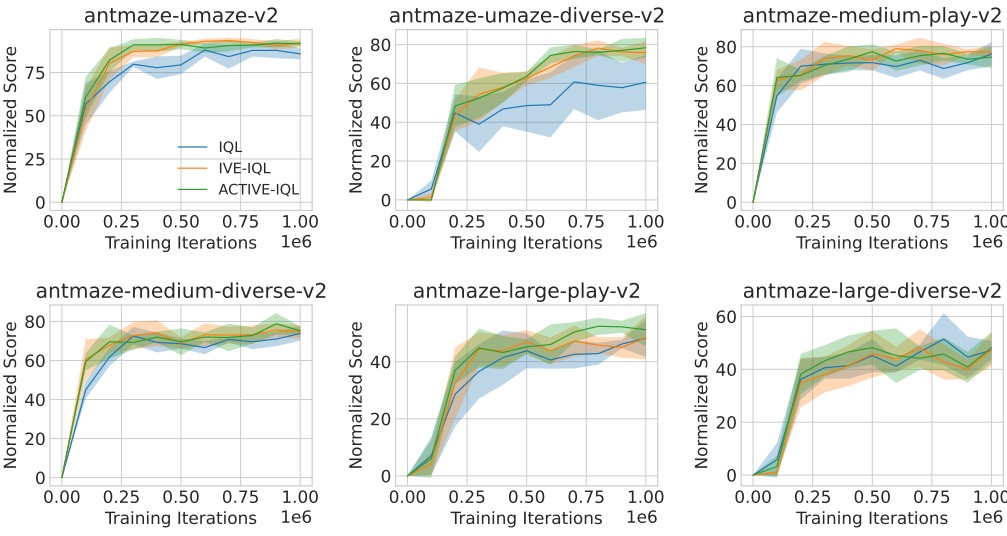

Figure 8: Learning curves of ACTIVE-I, IVE-I and IQL on AntMaze datasets.

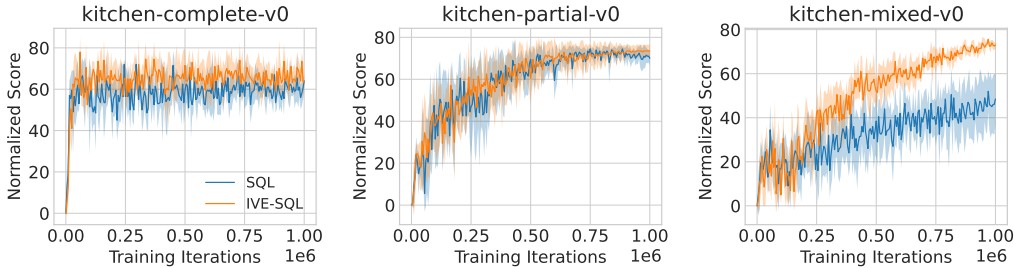

Figure 9: Learning curves of IVE-S and SQL on Kitchen datasets.

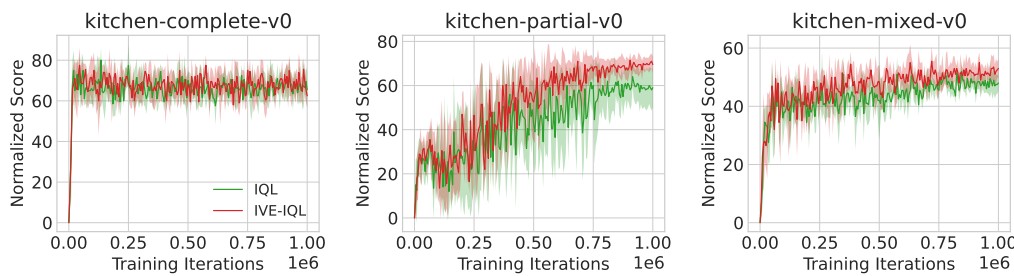

Figure 10: Learning curves of IVE-I and IQL on Kitchen datasets.

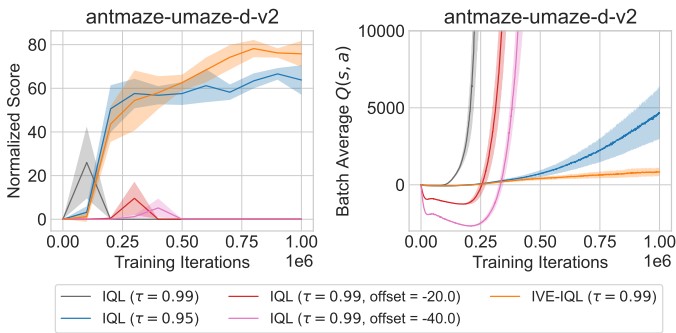

Figure 11: Normalized score and batch average $Q(s, a)$ of IQL and IVE-IQL with varied $\tau$. Additionally, we train IQL agents with $r(s, a) + \gamma V(s') + \text{offset}$ as the $Q$-target. We see that a constant penalty is not enough to suppress error accumulation and amplification.

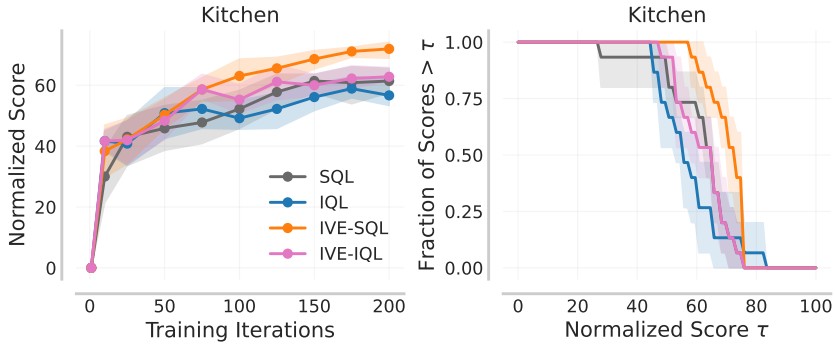

Figure 12: Average learning curves with stratified bootstrap confidence intervals (CIs) and performance profiles generated by the rliable library (Agarwal et al., 2021) on Kitchen tasks.

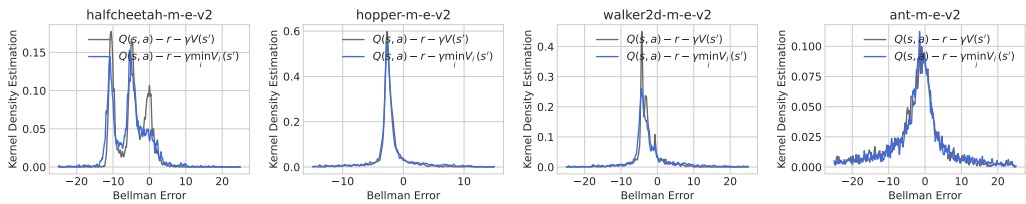

Figure 13: Kernel density estimation of Bellman errors obtained by training on 4 datasets from the D4RL benchmark using single network SQL (gray) and $V$-ensemble (blue).

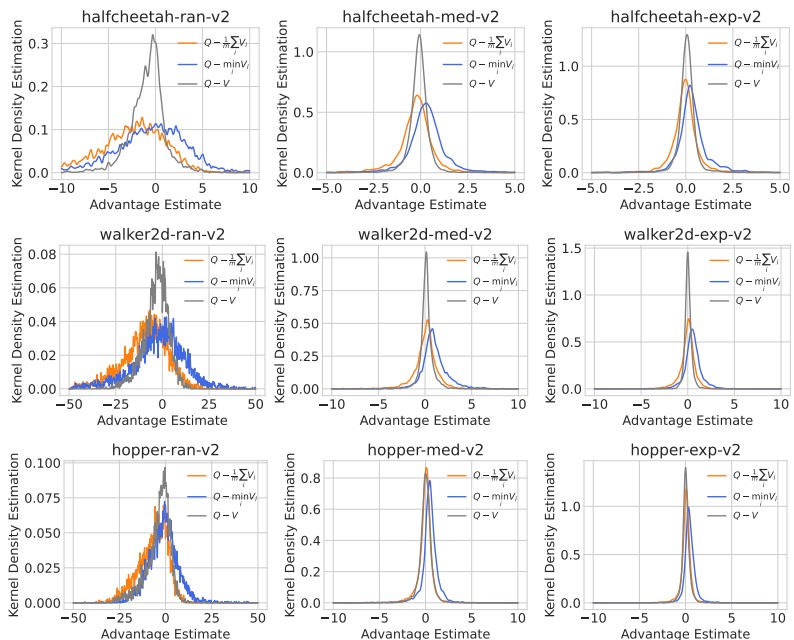

Figure 14: Kernel density estimation of the advantage distribution on 12 datasets from the D4RL benchmark obtained using single network SQL (gray) and $V$-ensemble (blue and orange).

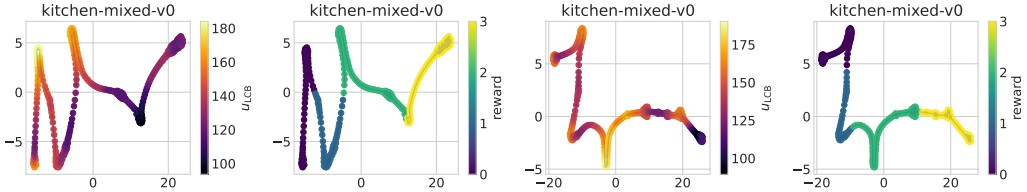

Figure 15: Uncertainty estimate $\mathrm{mean}_i V_i - \mathrm{LCB}(V_i)$ and reward visualization using t-SNE (van der Maaten & Hinton, 2008) projections of observations in Kitchen trajectories.

Table 8: Comparison against SAC-N on Kitchen tasks.

| Dataset | SAC-N ($m = 20$) | SAC-N ($m = 40$) | ACTIVE-S |
|---|---|---|---|
| kitchen-c | $0.0 \pm 0.0$ | $0.0 \pm 0.0$ | $66.5 \pm 3.3$ |
| kitchen-p | $0.0 \pm 0.0$ | $0.0 \pm 0.0$ | $73.5 \pm 3.0$ |
| kitchen-m | $0.0 \pm 0.0$ | $0.0 \pm 0.0$ | $73.2 \pm 1.4$ |

