# OpenReview forum: "ACTIVE: Offline Reinforcement Learning via Adaptive Imitation and In-sample $V$-Ensemble"
_ICLR.cc/2025/Conference — ICLR 2025 Poster_

### Official Review · Reviewer_oLG4 · 2024-10-28

**Soundness:** 2
**Presentation:** 3
**Contribution:** 2
**Rating:** 6
**Confidence:** 5

**Summary:**

This paper proposes an offline reinforcement learning algorithm ACTIVE that can mitigate the over-regularization problem of existing in-sample learning schemes. This paper first demonstrates through experiments that using a higher $\tau$ in IQL (or using a lower $\alpha$ in SQL) will lead to unstable estimates during the training process. And they demonstrate theoretically that V-ensemble can mitigate value overestimation by suppressing the accumulation of initial value errors, and experimentally demonstrates that ACTIVE can mitigate the overfitting of the value function and thus outperforms existing in-sample methods.

I have reviewed this manuscript submitted to NeurIPS 2024, and the authors have modified the manuscript in response to my comments and added comparison experiments with some ensemble algorithms, and these modifications address most of my questions.

**Strengths:**

1. This method has reasonable motivation and theoretical basis.
2. This method could help address the issue of in-sample value learning and policy extraction methods performing poorly on suboptimal or compositional datasets.
3. The experiments demonstrate that the method achieves some empirical improvements.

**Weaknesses:**

1. The innovation of this paper is moderate. As far as I know, the use of ensemble tricks to stabilize value estimation has been widely used in the RL field.
2. The algorithm introduces two new superparameters $c$ and $\mathcal{H}$, which are somewhat sensitive according to Figure 4 and Figure 5. Especially $\mathcal{H}$, it needs to be adjusted for specific datasets.

**Questions:**

1. Besides the run time, can you provide a comparison of the algorithm's parameters?

---

> ### Author Response · Authors · 2024-11-15
>
> We appreciate your time to review and your continued recognition of our work!
>
> > Besides the run time, can you provide a comparison of the algorithm's parameters?
>
>  As an example, consider halfcheetah-medium-v2 in which the observation dimension $d = 17$. A single $V$-network, which is a 3-layer ReLU activated MLP with 256 units, has 70,657 parameters. In our experiments, a $V$-ensemble typically has size $m = 5$, which amounts to 353,285 parameters.

---

> > ### Comment · Reviewer_oLG4 · 2024-11-19
> >
> > Thank you for your response. After reviewing your response and other reviews, I will keep my score.

---

### Official Review · Reviewer_L27K · 2024-11-01

**Soundness:** 2
**Presentation:** 3
**Contribution:** 2
**Rating:** 6
**Confidence:** 4

**Summary:**

This paper addresses the observed overfitting issue of V-functions in offline reinforcement learning. Specifically, the authors claim that a V-function may overfit to unstable overestimates in the target Q-values, which can, in turn, adversely impact the learning of both the actor and critic. The authors propose two primary techniques to mitigate this issue:

**First Technique: IVE (V-function Ensemble and Pessimistic Estimation).** To mitigate overfitting and prevent exploding Q-values, the authors propose maintaining an ensemble of V-functions and using their minimal estimate as the target during policy improvement:

$$L\_V^f(\psi\_i) = \mathbb{E}\_{(s,a)\sim D} [f(Q\_{\hat{\theta}} - V\_{\psi\_i}(s))]$$

$$L\_Q(\theta) = \mathbb{E}\_{(s,a,s')\sim D} [ \Big(r(s,a) + \gamma \min\_i V\_{\psi_i}(s') - Q\_\theta(s,a)\Big)^2]$$

, where $f$ is a general distance function. Seems the author use $\alpha$-divergence, as implied in Algorithm 1 (ACTIVE) on page 6.

The intuition behind this V-function ensemble is that for unstable or highly overestimated target Q-values, there is likely to be at least one V-function instance in the ensemble that does not fit this overestimated target well. This imperfect approximation helps prevent catastrophic overestimation.

**Secondly, ACT (Adaptive Cloning Temperature)**

The authors propose an adjustment to the advantage function used in policy extraction, building on modifications to the vanilla Advantage-Weighted Regression used in SQL and IQL:

$$\hat{A}(s,a) := Q\_{\hat{\theta}}(s,a) - \mathbb{E}^c \[V\_{\psi_i}\](s)$$

with $\mathbb{E}^c$ denotes the c-th quantile of the V-predictions. Besides, the author conduct policy extraction on both Q-function and A-function, which is different to IQL and SQL.

**Strengths:**

1. This paper is well-organized. Presenting work with two distinct techniques can be challenging, yet this paper does so effectively. The descriptions of each technique are clear, with a justification provided for the first technique, IVE.

2. It is a noteworthy finding that sota offline RL algorithms, such as SQL and IQL, still encounter overestimation issues in certain scenario. The introduction of a V-function ensemble to mitigate this problem yields incremental improvements, as demonstrated in Table 2 of the experiments. I also observed that the reproduced IQL results are slightly better than those in the original paper, while the SQL results are slightly lower—both of which align with expectations based on my own experience reproducing these algorithms in my research.

**Weaknesses:**

**(Minor)** **Inconsistent naming of techniques**. The names of the proposed techniques are inconsistent, which may confuse readers. The second technique, ACT, is mentioned under various names, including *automating likelihood adjustment*, *adaptive cloning temperature*, and *adaptive imitation*. Maintaining a single, consistent name for each technique would improve clarity.

**(Minor) Insufficient theoretical analysis for V-function ensemble.** The theoretical analysis supporting the V-function ensemble feels somewhat lacking. My understanding is that the theorem attempts to show that recursively expanding the pessimistic target could provide certain benefits:
$$Q = [1 + \gamma + ... + \gamma^t]r + (\Omega - B) \gamma [1 + \gamma + ... + \gamma^t] + \mathcal{O}$$
where B is some form of $V_\text{var}$, and $\Omega$ is a form of initial error. I believe this is not entirely new to the RL community. Additionally, the analysis lacks a convergence guarantee to ensure that such a recursive expansion can be reliably conducted.  Besides, the assumption 4.3 that $\pi_\text{imp} \propto \dfrac{|f'(Q(s,a) - V^*(s))|}{|Q(s,a) - V^*(s)|}$ is fixed throughout the learning process, seems infeasible. Since we are continuously updating the Q-function during Q-learning, it would be challenging to hold this assumption constant.

**Questions:**

I tend to rate this paper as “marginally below the acceptance threshold” for the following reasons:

1. **V-function ensemble in offline RL**: Introducing a V-function ensemble for offline RL is somewhat interesting. Although V-function ensembles are new to me, they appear conceptually similar to existing Q-function ensembles, such as REM and other methods with pessimistic estimates (e.g., SAC-N, PBRL, EDAC, VPQ).
2. **Insight on overestimation in sota Offline RL**: This paper makes an interesting observation that sota offline RL algorithms still encounter overestimation issues in some scenarios. While the authors refer to this as “overfitting,” it is, in effect, overfitting to unstable overestimated Q-values.
3. **Clarity of contributions.** The contributions of this paper feel somewhat overshadowed by an abundance of technical details. The modifications proposed by this paper include:
    1. *Equations (6) and (7)*: A V-function ensemble designed to approximate stable Q-targets while avoiding unstable overestimated values,
    2. *Equation (6)*: Using a suitable  f-divergence for mapping the V-function to the Q-value of the best action in the dataset, and
    3. *Equations (14) and (15)*: Extracting the policy based on the learned Q-function, V-function, and a quantile.

    Each of these components would benefit from comprehensive analysis, but in this paper, they are presented in a tightly condensed manner aimed at deriving a sota offline RL algorithm. While I respect the author’s effort, this approach makes it challenging to appreciate each individual contribution.

In summary, I find the core idea of this paper less compelling, with limited new insights or utility in the analysis. The primary contribution appears to be the development of a potentially strong offline RL algorithm, rather than a well-motivated solution for offline RL. Nevertheless, I rate this paper “marginally below the acceptance threshold” to avoid letting my personal preferences bias the review of this well-executed experimental study.

Question for the author: I believe Equation (8) may be unnecessary, as it could potentially confuse readers, especially when read alongside Equations (14) and (15). Right?

---

> ### Author Response · Authors · 2024-11-16
>
> We appreciate your time to review and your recognition of our experiments.
>
> > Maintaining a single, consistent name for each technique would improve clarity.
>
> To ensure naming consistency, we have revised the manuscript and changed the title of Section 5 to "Policy Extraction with Adaptive Cloning Temperature".
>
> > this approach makes it challenging to appreciate each individual contribution.
>
> In the last two parts of Section 7.1, each of the two methods, ACT and IVE, are evaluated on their own. The results shown in Figure 3 and 4 are obtained using IVE only. On the other hand, the results shown in Figure 5 are obtained using ACT only. The caption of Figure 5 has been corrected in the latest manuscript by replacing "ACTIVE-SQL" with "ACT-S".
>
> Thank you for your thorough and detailed review. We hope that our corrections and clarifications address the concerns you've raised. Please let us know if there are other issues we can address that would enable you to increase your score for the paper.

---

> > ### Comment · Reviewer_L27K · 2024-11-19
> >
> > Dear Author and AC,
> >
> > Thank you for your revision. It has significantly improved the readability of the paper, and I appreciate the effort made to address my earlier comments.
> >
> > I acknowledge the authors’ clarification regarding the ablation study on ACT and IVE. However, my main concern remains unresolved. Specifically, IVE (the V-function ensemble and the lower-bound target) is proposed to address the overestimation issue in Q-value estimation. The addition of ACT (adaptive cloning temperature and a new loss function for policy extraction) feels somewhat redundant or unclear in this context. If IVE is effective in mitigating overestimation, what is the necessity of introducing ACT?
> >
> > I believe that the inclusion of ACT is primarily aimed at outperforming these existing sota offline RL methods. While being the best-performing offline RL method is impressive, I think it’s even more important to understand why these methods work the way they do. For example, it would be helpful to explore why V-functions overfit to overestimated Q-values and how the ensemble of V-functions helps. A simple toy example to show the overestimation problem would also help make the motivation clearer, rather than just relying on performance results / Q-value on the D4RL datasets (say, Figure 1 is not very convincing because there are many parameters that make the Q-function explode).
> >
> > **In conclusion, I am inclined to maintain my original score.** I acknowledge that **my comments reflect a lot of personal preference**, and I trust that the ACs will take this into account to ensure a balanced and fair decision.
> >
> > Thanks. Please correct me if you find mistakes in my comment.

---

> ### Author Response · Authors · 2024-11-20
>
> Thank you for your follow-up and for carefully considering our response. We believe there may be a misunderstanding about our method, and we will try to clarify these points in the paper.
>
> > If IVE is effective in mitigating overestimation, what is the necessity of introducing ACT?
>
> The primary goal of this paper is mitigating over-regularization of **policies** learned using in-sample algorithms, which could be caused by the following reasons:
> 1. **Insufficient approximation of the in-sample max**: In theory, $\tau$ should be close to 1 to obtain a near-optimal policy. However, in practice a larger $\tau$ may give a worse result. In this paper, we identified overestimation as the main source of this subproblem and proposed IVE as a remedy.
> 2. **Inherent conservatism of AWR**: This problem cannot be fixed by value function modifications only. The idea of controlling the target likelihood $\mathcal{H}$ allows us to examine how close the policy is to the dataset. Empirical results, as shown in Figure 5, indicate that being too close to the dataset is indeed suboptimal, since moving slightly away from the dataset does improve performance.
>
> Therefore, mitigating overestimation is not the ultimate goal in this paper. This misunderstanding is similar to what reviewer wzeE had mentioned in the initial review. Policy over-regularization could still happen even if we controlled value overestimation.
>
> > (say, Figure 1 is not very convincing because there are many parameters that make the Q-function explode)
>
> Please check the results shown in Figure 11 (as well as [link1](https://ibb.co/Mf7F9YF), [link2](https://ibb.co/jkvpm0S) on NeoRL datasets), in which we tuned the parameter $\tau$ and it correlates well with overestimation across domains.
>
> > For example, it would be helpful to explore why V-functions overfit to overestimated Q-values and how the ensemble of V-functions helps.
>
> Theorem 4.5 provides a possible explanation of a simplified version of the problem. Briefly speaking, the Q-error is amplified by $\pi_{imp}$ (which is optimized using the V-loss in practice) and accumulated during in-sample TD updates. This process is stabilized with ensemble-based LCB updates. Note that our NTK-based analysis operates on data matrices. The $B$ in Equation 10 is actually a function of transitions $(s, a, r, s')$ where different rows corresponds to different transitions. Therefore, $B$ is adaptive in the sense that the strength of penalty can vary across states based on the uncertainty captured by the ensemble.
>
> Thank you for your thorough and detailed review. We remain committed to further refining the manuscript. We hope that our corrections and clarifications address the concerns you've raised. Please let us know if there are other issues we can address that would enable you to increase your score for the paper.

---

> > ### Comment · Reviewer_L27K · 2024-11-23
> >
> > It is new to me that reverse KL divergence is considered a form of conservatism. To facilitate the discussion, I will provide some background information to ensure we are on the same page.
> >
> > (1) the optimal policy defined in AWR. AWR concerns the analytic solution for the constrained optimization problem: $\arg\max_{\pi} \mathbb{E}_{a\sim \pi(\cdot |s)}[A^*(s,a)]$, with reverse KL constrain $D\_{\text{KL}}(\pi(\cdot | s) | \pi\_\beta) < \epsilon$. Here, $\pi\_\beta$  is the behavior policy that collects the dataset and $A^*$ is the optimal A-function (and is nothing with the solution or $\pi\_\beta$). Denote the solution to this optimization problem as $\pi^*$, then $\pi^*(\cdot | s)= \frac{1}{Z(s)} \pi\_\beta(\cdot |s) \exp(\frac{1}{\lambda}A^*(s,a))$.
> >
> > (2) To approximate $\pi^*$ using a parameterized policy $\pi\_{\theta}$,  AWR needs forward KL: $\arg\min\_{\pi\_\theta} KL(\pi^* | \pi_\theta(\cdot | s))$. This leads to the optimization objective used in AWR or IQL: $\arg\max\_{\pi\_\theta} \mathbb{E}\_{s \sim d} \mathbb{E}_{a \sim \pi\_\beta(\cdot |s)} \log \pi\_\theta(\cdot |s) \cdot \exp(\frac{1}{\lambda} A^*(s,a))$.
> >
> > In summary, the process involves first applying a reverse KL divergence (which encourages mode-seeking behavior) and then a forward KL divergence (which encourages mean-seeking behavior).
> >
> >
> > The forward KL is used to constrain the learned policy twoard $\pi^*$. I agree that, I agree that, if $\pi^*$ is a good solution, the forward KL can work well. Note here $\pi^*$ is not the optimal policy regarding the return-it is just-a solution of the defined objective.
> >
> > The reverse KL, on the other hand, is used to define the objective and the solution itself. However, the optimality of the solution $\pi^*$, I believe, is missing (I carefully checked the relevant AWR and AWAC papers). Since you argue that the reverse KL is conservatism. I believe it would be stronger to support your claim with concrete evidence rather than simply asserting that "reverse KL is conservatism." At least two approaches could help support this claim:
> >
> >
> > You can compare the regret of the objective with reverse KL to that of ACT. This could provide a theoretical justification.
> >
> > Or, you can obtain a optimal A-function using some online RL algorithm (have both V and Q) and then extract policies using replay dataset and the two objectives, and compare their performance.
> >
> > What do you think? If you believe this verification is unnecessary, please let me know.

---

> ### Author Response · Authors · 2024-11-23
>
> Thank you for your follow-up and for carefully considering our response. We apologise for not stating the problem correctly in the previous comment.
>
> First, we have the optimal (implicit) policy defined in AWR:
> $$\pi^{\star}(a | s) \propto \mu(a|s)\cdot \exp(A^{\star}(s, a) / \lambda)$$
> which is indeed obtained using reverse KL $D_{\text{KL}}(\pi | \mu)$ between $\pi$ and $\mu$.
>
> Now during the **policy extraction** phase, we consider the reverse and forward KL **between $\pi_{\theta}$ and $\pi^{*}$**. We refer to the discussion in Extreme Q-learning ([1], Section 3.4), in which the forward KL corresponds to an AWR loss, and the reverse KL corresponds to a SAC-like loss.
>
> Forward KL (AWR): $\mathbb{E}_{(s, a)\sim \mathcal{D}}[\exp(A(s, a) / \lambda) \log \pi(a | s)]$
>
> Reverse KL (SAC-like): $\mathbb{E}_{s \sim \mathcal{D}, a \sim \pi(\cdot|s)} \left[ Q(s, a) - \lambda \log\left( \frac{\pi(a|s)}{\mu(a|s)} \right) \right]$
>
> From looking at the loss functions, the forward KL is **fully in-sample**, which means that it only querys values of dataset actions and adjust their probabilities $\log \pi(a | s)$. Furthermore, it means that **at least some** actions in the dataset would increase in probability. On the other hand, the reverse KL can adjust the probability of actions $a \sim \pi(\cdot|s)$ but has to be regularized.
>
> In multiple instances of our experiments (e.g., halfcheetah-m, halfcheetah-m-r), AWR tends to **monotonically** increase dataset likelihood, which is good for regularization at the beginning but can be unnecessary for the last 80% of total iterations. We therefore hypothesize that **AWR/forward KL** could be a source of over-regularization. This observation led to the design of ACT.
>
> Thank you for your continued engagement! We hope that our corrections and clarifications address the concerns you've raised. Please let us know if there are other issues we can address.
>
> ---
>
> [1] Garg, D., Hejna, J., Geist, M., Ermon, S., 2023. Extreme Q-Learning: MaxEnt RL without Entropy, in: International Conference on Learning Representations.

---

> > ### Comment · Reviewer_L27K · 2024-11-23
> >
> > You are correct; the empirical results in Figure 5 provide strong motivation for ACT. I have no further questions or concerns at this time and will therefore increase my score from 5 to 6. I will provide detailed feedback later.

---

> > > ### Author Response · Authors · 2024-11-23
> > >
> > > Thank you for raising the score and actively participating in the discussion. We greatly appreciate your time and effort!

---

### Official Review · Reviewer_wzeE · 2024-11-02

**Soundness:** 2
**Presentation:** 2
**Contribution:** 2
**Rating:** 6
**Confidence:** 3

**Summary:**

The paper focuses on resolving the problem of overregularization induced by in-sample learning methods in an offline RL setting. It provides a motivating experiment to demonstrate that existing approaches like IQL could still suffer from overestimation in the late learning stages. The proposed method differs from existing IQL algorithms mainly in two ways: 1) it introduces an ensemble to learn the value function; and 2) it introduces an entropy constraint on the policy learning, which is further reformulated by Lagrangian and solved by dual gradient descent. Empirical results on commonly used offline RL datasets are presented to verify the effectiveness of the proposed algorithm.

**Strengths:**

1. The paper studies an important topic.
2. The paper presents promising empirical results.
3. The paper explains its main algorithm clearly.

**Weaknesses:**

The main weaknesses are as follows.

Definition of Over-Regularization: The paper repeatedly emphasizes that existing approaches suffer from over-regularization induced by in-sample constraints, while the proposed method can mitigate this. However, the term "over-regularization" is never clearly defined. Intuitively, it might imply that the learned action values are somewhat underestimated due to in-sample constraints. Yet, it is confusing that Section 4.1 claims that existing algorithms like SQL still overestimate. If overestimation is an issue, doesn’t this suggest that there is insufficient regularization? And, isn't it simpler to bring in methods to mitigate overestimation (there were already many methods proposed for this)?

Theoretical Clarity: The theoretical result appears confusing. Theorem 4.5 states that the initial error can be penalized, but it is unclear how this penalization of initial error relates to preventing overestimation.

Choice of Competitors for Empirical Testing: Figure 1 omits tests on the original IQL and CQL. Additionally, a fair comparison seems to be missing; the proposed approach should ideally be compared with in-sample softmax by Chenjun Xiao et al., as cited by the authors. This paper directly bootstraps from in-sample actions and also incorporates entropy regularization, and it was proved to converge to an optimal entropy-regularized policy on the empirical MDP defined by offline dataset.

Effectiveness of the Proposed Method: Line 286 states, "...but cannot completely solve the overregularization problem." However, it remains questionable whether even with the proposed dual gradient descent method, this problem can be fully resolved.

Need for Ablation Study: The paper should include an ablation study on the use of the ensemble value function and dual gradient descent to carefully verify their effect and necessity.

**Questions:**

see above.

---

> ### Author Response · Authors · 2024-11-16
>
> Thank you for your comments and feedback. We believe there may be a misunderstanding about our method, and we will try to clarify these points in the paper.
>
> **"Need for Ablation Study"**
>
> 1. The paper does include a dedicated section for ablation study (Section 7.2). Aggregated results are shown in Figures 6 and 12.
> 2. Each of the two methods (IVE and ACT) is evaluated independently in Section 7.1. The results are shown in Figures 3-5.
> 3. More results on individual datasets are shown in Figures 7-10.
>
> **"Definition of Over-Regularization"**
>
> 1. The term "over-regularization" is loosely defined as the phenomenon of policies being skewed toward suboptimal behavior due to policy/value constraints. This is also the case in similar studies (e.g. PRDC [1]). The reason is that the policy degradation has several different sources, which might include over-conservative value estimates and the mean-seeking behavior of AWR [2]. In this work, we decompose the notion of "over-regularization" into two specific problems, including (i) the inability to choose a lower constraint $\alpha$ due to overestimation and (ii) the mean-seeking behavior of AWR. These two problems are alleviated by the proposed IVE and ACT methods, respectively.
> 2. Note that AWR only depends on the *difference* between $Q(s, a)$ and $V(s)$. Thus, even if the value function is overestimated, the policy may still be constrained to suboptimal behavior if $A(s, a) = Q(s, a) - V(s)$ does not filter actions correctly.
>
> **"Choice of Competitors for Empirical Testing"**
>
> 1. Among similar in-sample offline RL algorithms, including IQL, SQL and InAC (in-sample softmax [3]), SQL typically achieves the highest scores on D4RL. We think that it is reasonable to choose SQL as the main in-sample competitor.
> 2. A similar overestimation effect can be observed in IQL, as shown in Figure 11.
>
> **"Effectiveness of the Proposed Method"**
>
> We do not claim that the problem can be fully resolved with the proposed method. However, the ACT approach provides performance increases in some cases that would be difficult to achieve by IVE alone (Figure 7).
>
> **"Theoretical Clarity"**
>
> In Theorem 4.5, we calculated the $Q$-value estimate on the dataset after $(t+1)$ iterations of in-sample updates. Computing the exact value would be difficult, since the result depends on the specific dataset $\mathcal{X}$ and the network architecture. We present empirical evaluation results in Figure 11, which shows that the IVE technique successfully mitigates overestimation.
>
> Thank you for your review. We hope that our corrections and clarifications address the concerns you've raised. Please let us know if there are other issues we can address that would enable you to increase your score for the paper.
>
> [1] Y. Ran, Y.-C. Li, F. Zhang, Z. Zhang, and Y. Yu, “Policy Regularization with Dataset Constraint for Offline Reinforcement Learning,” in _International Conference on Machine Learning_, PMLR, 2023, pp. 28701–28717. Available: [http://arxiv.org/abs/2306.06569](http://arxiv.org/abs/2306.06569)
>
> [2] D. Garg, J. Hejna, M. Geist, and S. Ermon, “Extreme Q-Learning: MaxEnt RL without Entropy,” in _International Conference on Learning Representations_, 2023. Available: [http://arxiv.org/abs/2301.02328](http://arxiv.org/abs/2301.02328)
>
> [3] C. Xiao, H. Wang, Y. Pan, A. White, and M. White, “The In-Sample Softmax for Offline Reinforcement Learning,” Feb. 28, 2023, _arXiv_: arXiv:2302.14372. Available: [http://arxiv.org/abs/2302.14372](http://arxiv.org/abs/2302.14372)

---

> ### Author Response · Authors · 2024-11-23
> **Additional clarifications on the choice of competitors**
>
> From Equation (15) in the paper of in-sample softmax [1], we can see that the in-sample softmax algorithm queries actions from the current policy $\pi_{\psi}$:
>
> $$L_{\text{baseline}}(\phi) = E_{s \sim D, a \sim \pi_\psi(s)} \left[ \frac{1}{2}(v_{\phi}(s) - (q_{\theta}(s, a) - \tau \log \pi_{\psi}(a | s)))^{2} \right]$$
>
> However, in our paper we only consider algorithms that fall under the generalized IQL framework, which involves a potentially asymmetric loss $f$ and does not query actions outside the dataset **during bootstrapping**. Therefore, the in-sample softmax algorithm was not chosen as a competitor in our experiments.
>
> Thank you for your review. Please let us know if any further questions remain. We hope the reviewer can reassess our work with these clarifications.
>
> ---
>
> [1] C. Xiao, H. Wang, Y. Pan, A. White, and M. White, “The In-Sample Softmax for Offline Reinforcement Learning,” in _International Conference on Learning Representations_, 2023.

---

> ### Author Response · Authors · 2024-11-24
> **Request for Further Review Feedback**
>
> Dear Reviewer,
>
> As the end of the discussion period approaches, we would greatly appreciate it if you could confirm whether our response has adequately addressed your concerns. If you have any remaining questions, please let us know, and we will do our best to respond within the remaining time.
>
> If your questions have been addressed, we kindly ask you to consider raising your rating. Thank you again for your time and efforts in reviewing our manuscript.
>
> Sincerely,
>
> The Authors

---

### Official Review · Reviewer_J6er · 2024-11-04

**Soundness:** 3
**Presentation:** 3
**Contribution:** 2
**Rating:** 6
**Confidence:** 3

**Summary:**

This paper aims to address the overestimation problem of offline RL methods that learn in-sample value functions to avoid querying out-of-distribution samples. The main idea is (i) to learn ensemble of V functions and use clipped V function for learning V functions and (ii) use automatic tuning of hyperparameter used for in-sample value learning such as quantile regression. Arguments for introducing the idea are strong. Several ablation/analysis are provided to support the claims in the paper. Experiments are conducted on D4RL benchmark.

**Strengths:**

The main strength of this paper is that the paper does not have any big flaw and does not make any big mistake. Writing is clear, motivation is clear, intuition is clear, and the results look okay. The idea of learning ensemble of value functions has been very popular in offline RL context and also clipped Q learning is also widely-used, so the idea of using it for regularizing IQL-like method is solid.

**Weaknesses:**

- The main problem of this paper is that it's not clear how important the problem this paper aims to address, or the importance of such problem is not clearly evidenced & presented in the writing, mainly because D4RL is quite saturated benchmark at this point and it's not clear how the observation / results in this benchmark would transfer to other scenarios.
- It's not clear the motivating overfitting problem can be generally observed in diverse domains or it is only happening on D4RL tasks. Motivating figures could be provided on more tasks / domains to support that overfitting is a common problem across various scenarios. More results on different tasks from different benchmarks such as NeoRL or visual offline RL benchmarks can be a plus.
- Weak performance compared to MSG is not clearly justified to me; if that is from not using CQL-like regularizer, can ACTIVE be better if it incorporates similar regularizer?
- It could be nice to make tables be more self-contained as far as possible. For instance, make it clear that what -I and -S means in each caption, or make it clear what $m$ means in Table 1.

**Questions:**

See Weaknesses.

---

> ### Author Response · Authors · 2024-11-16
>
> Thank you for your comments and feedback.
>
> > For instance, make it clear that what -I and -S means in each caption, or make it clear what $m$ means in Table 1.
>
> We have revised the captions of Table 1 and 2 accordingly.
> - (Table 1) Comparison against ensemble-based algorithms with similar ensembles sizes ($m$).
> - (Table 2) The "-I" and "-S" suffixes indicate the use of IQL-style and SQL-style $V$-function losses, respectively.

---

> ### Author Response · Authors · 2024-11-19
> **Additional results**
>
> > More results on different tasks from different benchmarks such as NeoRL or visual offline RL benchmarks can be a plus.
>
> We provide additional results on the NeoRL datasets. The results, available at [link1](https://ibb.co/Mf7F9YF), [link2](https://ibb.co/jkvpm0S), show that overfitting is generally observed across various scenarios. We are working on a full benchmark on 9 NeoRL tasks.
>
> It is known that the hyperparameter $\tau$ has a gap between theory and practice: in theory $\tau$ should be close to 1 to obtain an optimal policy while in practice a larger $\tau$ may give a worse result or lead to numerical instabilities [1]. In our work, IVE better bridges the gap by controlling overestimation and allowing a larger $\tau$.
>
> Please let us know if any further questions remain. We hope the reviewer can reassess our work with these clarifications.
>
> ---
>
> [1] H. Xu _et al._, “Offline RL with No OOD Actions: In-Sample Learning via Implicit Value Regularization,” in _International Conference on Learning Representations_, 2023.

---

> > ### Comment · Reviewer_J6er · 2024-11-25
> >
> > Thank you for your response. I acknowledge that I have read other reivews and the response. The response is not answering my question on the reason why the proposed mehtod is worse than MSG, which is not a big issue for me but it's still disappointing. As other reviewers also pointed out, I still think the contribution of this paper is a bit limited to warrant a score increase. I'm still on a very borderline between the score of 5 and 6 and will finalize the score after all the discussion phases.

---

> > > ### Author Response · Authors · 2024-11-25
> > >
> > > Thank you for your feedback and for considering the discussion phases before finalizing your score. We deeply appreciate the significant amount of time and effort you have dedicated to reviewing our manuscript. Your thorough and insightful feedback has been instrumental in guiding us to make meaningful improvements. We apologize if our previous response did not address your question to your satisfaction.
> > >
> > > To clarify, the proposed method may appear less effective than MSG on certain datasets, which we acknowledge (and mentioned in Section 8). However, our intention was to highlight its advantages on suboptimal/mixed datasets like **kitchen-mixed**, on which SAC-N (another prior ensemble-based algorithm) failed to produce meaningful policies while the proposed method enjoys a significant performance boost over in-sample baselines (Tables 2 and 8). The paper mainly focuses on **improving in-sample algorithms**. Still, we believe that the performance discrepancies between ACTIVE and MSG may have revealed a limitation of in-sample updates, which could be an area of further study.
> > >
> > > As for the question on incorporating a CQL-like regularizer:
> > > - To the best of my knowledge, CQL was designed to target OOD actions. However, in-sample algorithms avoid querying values of OOD actions. Therefore, our method might be addressing a different problem.
> > > - The IVE method already introduces a degree of pessimism. Therefore, incorporating CQL as an additional source of pessimism may be less well motivated.
> > >
> > > We are committed to continuous improvement and further exploration of the issues raised. We believe that the value of our method on suboptimal/mixed datasets will be a useful point of reference for the community. Thank you once again for your invaluable feedback and for helping us to refine our work.

---

### Meta-Review · Area_Chair_hu5a · 2024-12-22

**Metareview:**

The paper describes a new technique for offline RL.  The new technique uses a ensemble of V-networks with temperature adjustment to mitigate over estimation.  The strengths of this paper are the effectiveness of the approach, the justification for the design choices and the theoretical analysis.  The main weakness is the incremental nature of the work since using an ensemble to mitigate overestimation is a common trick in RL.  Nevertheless, the approach makes sense, it is effective and it advances the state of the art in offline RL.

**Additional Comments On Reviewer Discussion:**

There was no additional discussion since the reviewers unanimously recommend acceptance.  The reviewers kept their score at "marginal accept" mostly due to the incremental nature of the work since using an ensemble to mitigate overestimation is a common trick in RL.

---

### Decision · Program_Chairs · 2025-01-22

Accept (Poster)